# Towards Efficient LLMs Annealing with Principled Sample Selection

Yuanjian Xu [1 2]   Jianing Hao [1]   Wanbo Zhang [3]   Zhong Li [2 *]   Guang Zhang [1 *]

## Abstract

The annealing phase is a pivotal convergence stage in LLM pre-training that ultimately determines final model quality. However, effectively selecting training data during this phase remains a key challenge. Current strategies rely on empirical heuristics, such as domain filtering or context extension, which lack a principled grounding in optimization theory. In this work, we characterize the annealing phase through the lens of the loss landscape's spectral geometry. We argue that optimal convergence requires gradient updates to satisfy heterogeneous constraints across different eigen-directions. Building on this insight, we formulate data selection as a problem of satisfying these directional constraints. To this end, we propose **DiReCT** (**Di**rectionally-**Re**strained **C**onstrained **T**raining), a novel framework that reformulates sample selection in the annealing stage as a constrained optimization problem. By imposing explicit directional constraints on per-sample gradients based on the spectral properties of the Hessian, **DiReCT** identifies samples that align with the optimal curvature-aware descent path. Extensive experiments across various model scales demonstrate that **DiReCT** consistently achieves state-of-the-art performance. For future research, code is available at https://github.com/xuyj233/Direct.

## 1. Introduction

While the capabilities of large language models (LLMs) are fundamentally determined by their training data (Xu et al., 2023; Luo et al., 2025a), effectively harnessing this data re-

quires sophisticated training strategies. Modern pre-training pipelines therefore commonly adopt a two-stage approach (Hu et al., 2024; Black et al., 2022; Grattafiori et al., 2024): a stable phase for acquiring foundational knowledge from massive datasets, followed by an annealing phase for final convergence and optimization. Crucially, this annealing phase relies on two key levers: learning rate decay and targeted data selection. However, while the former is well-studied, strategies for the latter remain largely empirical. This reliance on heuristics, absent a principled grounding in optimization theory, leads to sub-optimal data scheduling and inefficient resource allocation at this most critical juncture of training.

In the absence of a principled foundation, data selection in the annealing phase consequently revolves around two empirical heuristics. One focuses on capability refinement through the up-sampling of reasoning-heavy data (e.g., mathematics, code) (Grattafiori et al., 2024). The other prioritizes context extension by integrating long-sequence samples (Yang et al., 2025). In essence, both strategies induce targeted shifts in the data distribution. Although they yield gains, these heuristics remain disconnected from the underlying optimization dynamics, leading to suboptimal scheduling. A more principled approach would require a selection framework explicitly guided by the model's instantaneous optimization geometry. To design such a data selection strategy, we must first characterize how the annealing phase differs from stable training. Following the valley metaphor used in prior studies to characterize optimization (Wen et al., 2025), as shown in Figure 1, towards the end of stable training (Figure 1(a)), the optimization enters a regime where gradients oscillate within steep regions of the loss landscape with minimal effective descent, causing convergence to stagnate. Breaking this plateau therefore requires the annealing phase (Figure 1(b)) to enforce a dual objective: providing a strong vertical descent signal while constraining horizontal oscillations within a sharp subspace.

Driven by this geometric intuition, we introduce **DiReCT**, a framework that formalizes data selection through the lens of spectral geometry. Our analysis begins by characterizing the annealing-phase loss landscape via the Hessian matrix computed on a validation set. This reveals a fundamental tension between high-curvature ("sharp") directions, where optimization is prone to instability from gradient noise, and

---

Work done during Yuanjian Xu's internship at Microsoft Research Asia. [1]The Hong Kong University of Science and Technology (Guangzhou), Guangzhou, China [2]Microsoft Research Asia, Beijing, China [3]Fudan University, Shanghai, China. Correspondence to: Zhong Li <zhongli@microsoft.com>, Guang Zhang <guangzhang@hkust-gz.edu.cn>.

low-curvature ("flat") directions, which govern stable generalization. **DiReCT** resolves this tension by evaluating each training sample based on its gradient's projection onto these spectral components, actively selecting those that amplify the descent signal in flat valleys while suppressing noise in sharp directions. To render this principle scalable, we employ efficient randomized projections to approximate high-dimensional gradients, dramatically reducing computational overhead. Moreover, the selection criterion permits the entire data schedule to be precomputed with just a single forward pass over the dataset before training begins. Consequently, **DiReCT** seamlessly aligns sample selection with the underlying optimization dynamics, ensuring both efficient computation and stable convergence. Our contributions are summarized as follows:

- We propose **DiReCT**, to the best of our knowledge, the first systematic framework designed to guide data selection during the annealing phase of LLMs.

- **DiReCT** moves beyond empirical heuristics by leveraging the spectral properties of the Hessian matrix to guide the annealing phase. By identifying and addressing the tension between sharp directions and low-curvature valleys, **DiReCT** offers a robust mechanism to suppress optimization noise while maintaining a strong descent signal.

- We demonstrate the effectiveness of **DiReCT** through extensive experiments across diverse model architectures and scales. The results show that our approach consistently outperforms state-of-the-art empirical baselines in final model capabilities.

## 2. Related Work

We organize this section by reviewing data selection strategies across the two major stages of LLM pre-training: stable training and the annealing phase.

**Data strategies in stable training.** The initial phase of pre-training focuses primarily on the acquisition of fundamental linguistic structures. Although existing literature rarely distinguishes the stable training stage explicitly, most current data strategies are inherently designed for this period, focusing on balancing domain weights and prioritizing samples to ensure steady convergence. Current research has transitioned from manual heuristics to optimization-driven domain weighting. For instance, DoReMi (Xie et al., 2023) utilizes distributionally robust optimization to minimize excess loss, while REGMIX (Liu et al., 2025) and MixMin (Thudi et al., 2025) treat mixture identification as regression or convex optimization problems. Furthermore, recent works introduce adaptive scheduling to handle evolving data importance during this phase. Chameleon (Xie et al.,

2025) employs leverage scores for flexible reweighting, and Velocitune (Luo et al., 2025b) monitors the model's learning velocity to adjust sample weights in real-time. These strategies, often grounded in scaling law analyzes (Shukor et al., 2025), facilitate efficient convergence by ensuring the model maintains a steady learning signal throughout the massive-scale stable training process.

**Data Strategies in annealing phase.** The annealing phase transitions from stable training to convergence, shifting the objective from language acquisition to capability consolidation. As shown in Table 1, recent models upsample specific datasets during this stage to enhance performance. For instance, DeepSeek-V3, Qwen-3, and YuLan-Mini mix math, code, and science data to improve reasoning (DeepSeek-AI et al., 2025; Yang et al., 2025; Hu et al., 2024). Similarly, GLM-4, Kimi-1.5, and Llama-3 integrate long-context data to extend sequence processing (Zeng et al., 2024; Team et al., 2025; Grattafiori et al., 2024). However, these strategies remain empirical and lack a principled understanding of the selection mechanisms during this phase.

*Table 1.* A comparison of data prioritization strategies during the annealing phase across leading LLMs. Checkmarks (✓) indicate categories where specific upsampling or curriculum shifts have been reported.

| Model | Math | Code | Science | Long |
|---|---|---|---|---|
| Llama-2-L | | | | ✓ |
| Llama-3 | ✓ | ✓ | | |
| DeepSeek-V3 | ✓ | ✓ | ✓ | ✓ |
| Qwen-3 | | ✓ | ✓ | ✓ |
| Qwen-2.5-1M | | | | ✓ |
| Ministral-3 | | | | ✓ |
| YuLan-Mini | ✓ | ✓ | ✓ | ✓ |
| GLM-4 | | | | ✓ |
| Kimi-1.5 | ✓ | ✓ | | ✓ |

## 3. Preliminary Study

To establish a principled understanding of the annealing stage, we characterize it not merely as a learning rate decay , but as a critical transition in the model's spectral navigation of the loss landscape (Goyal et al., 2017; Wen et al., 2025). We build upon the empirical observation that neural loss landscapes exhibit a skewed eigenspectrum, with a minority of high-curvature directions and a vast, flat subspace. In this geometric view, the stable training phase can be seen as the model operating in a high-entropy regime where significant learning rates induce a large noise scale (Wu et al., 2020; Keskar et al., 2017). This noise, while providing a beneficial self-stabilization effect that repels the model from sharp minima, geometrically manifests as dominant updates along the high-curvature (stiff) subspace $\mathcal{S}_\perp$. These transverse components largely cancel out over iterations, yielding only a weak net signal for descending the valley floor (the

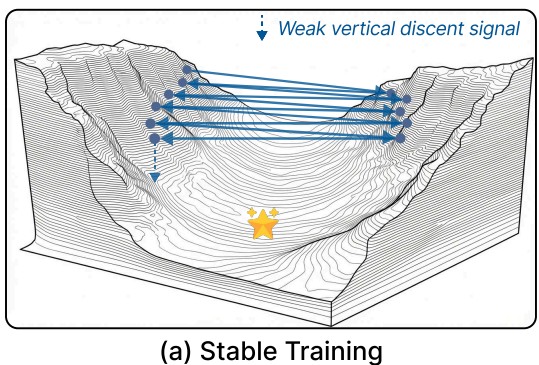

(a) Stable Training

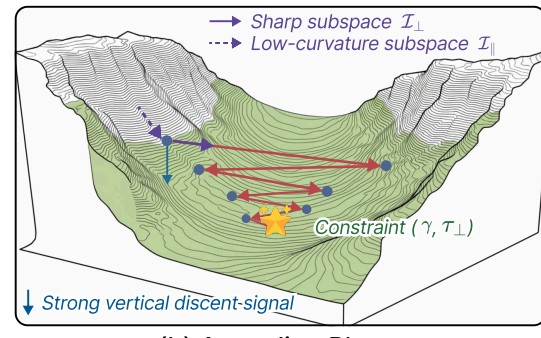

(b) Annealing Phase

*Figure 1.* A geometric comparison of optimization dynamics across training phases. In the stable training stage (a), high learning rates and uncurated data lead to high-variance oscillations across the steep walls of the loss landscape, resulting in a weak descent signal. During the annealing phase (b), our proposed mechanism suppresses transverse noise while maximizing the longitudinal signal, allowing the model to descend efficiently along the low-curvature valley toward the optimum.

low-curvature subspace $\mathcal{S}_{\parallel}$), consistent with observed slow descent prior to annealing.

The annealing phase, therefore, represents a targeted refinement process where the dynamics shift from exploration to exploitation of the low-curvature subspace. For this transition to be effective, the optimization must satisfy two distinct geometric conditions: mitigating noise-amplified perturbations in $\mathcal{S}_{\perp}$ to stabilize, while preserving a coherent descent signal in $\mathcal{S}_{\parallel}$ to accelerate along the flat valley floor. Our framework hypothesizes that the empirical success of certain training schedules stems from their implicit ability to satisfy these spectral constraints. In the following section, we formalize this property and actively control it through the design of a selection algorithm.

## 4. Methodology

We first establish the problem formulation in Section 4.1, defining the sample selection task. In Section 4.2, we introduce the construction of our optimization problem. We then derive the Successive Convex Approximation (SCA) solver in Section 4.3 to address the non-convex alignment constraints. Finally, Section 4.4 provides the implementation details, including the sketching-based approximation for high-dimensional Hessian analysis.

### 4.1. Problem Formulation

Let $D_{\text{train}} = \{x_i\}_{i=1}^{N}$ be the training dataset of text sequences, and let $\ell_{\text{pre}}(x; \theta)$ denote the self-supervised pretraining loss for a model with parameters $\theta \in \mathbb{R}^d$. Standard training minimizes the empirical risk $\mathcal{L}_{\text{ERM}}(\theta) = \frac{1}{N} \sum_{i=1}^{N} \ell_{\text{pre}}(x_i; \theta)$. We study post-hoc sample reweighting or selection during the late-phase training regime, where the model is near a local optimum. Let $w = (w_1, \ldots, w_N) \in \mathcal{W}$ be a vector of sample weights; the feasible set $\mathcal{W}$ encodes the chosen strategy: for reweighting, $\mathcal{W} \subseteq \mathbb{R}_{\geq 0}^{N}$ with a nor-

malization constraint, and for selection, $\mathcal{W} \subseteq \{0, 1\}^N$ with a cardinality constraint $\sum_i w_i = K$. Both modify the training objective to $\mathcal{L}_{\text{pre}}(\theta; w) = \frac{1}{\sum_i w_i} \sum_{i=1}^{N} w_i \, \ell_{\text{pre}}(x_i; \theta)$.

We assume access to a held-out validation set $D_{\text{val}} = \{x_j^{\text{val}}\}_{j=1}^{M}$ from the same pretraining distribution, used to guide weight optimization via the validation loss $\mathcal{L}_{\text{val}}(\theta) = \frac{1}{M} \sum_{j=1}^{M} \ell_{\text{pre}}(x_j^{\text{val}}; \theta)$. To evaluate downstream performance after pretraining, we use a supervised posttraining dataset $D_{\text{post}} = \{(x_k^{\text{post}}, y_k^{\text{post}})\}_{k=1}^{L}$, e.g., an supervised fine-tuning (SFT) benchmark, with loss $\ell_{\text{post}}(x, y; \theta)$.

Let $\theta_{\text{base}}$ denote the model obtained by standard ERM training. Starting from $\theta_{\text{base}}$, we perform a few additional training steps using the weighted loss $\mathcal{L}_{\text{pre}}(\theta; w)$ to obtain adjusted parameters $\theta_{\text{adj}}(w)$. Our goal is to maximize the post-training performance gain:

$$\max_{w \in \mathcal{W}} \left[ \mathcal{L}_{\text{post}}(\theta_{\text{base}}) - \mathcal{L}_{\text{post}}(\theta_{\text{adj}}(w)) \right], \quad (1)$$

where $\mathcal{L}_{\text{post}}(\theta) = \frac{1}{L} \sum_{k=1}^{L} \ell_{\text{post}}(x_k^{\text{post}}, y_k^{\text{post}}; \theta)$.

### 4.2. Annealing Data Selection as an Optimization

We begin by defining the model's parameter state $\theta_s \in \mathbb{R}^d$ at the end of stable training. To characterize the loss landscape geometry around $\theta_s$, we introduce a validation set $D_{\text{val}}$ representing the target data distribution and calculate the corresponding Hessian matrix as shown in Definition 4.1:

> **Definition 4.1** (Validation Hessian Matrix)**.** The validation Hessian matrix at $\theta_s$ is defined as:
>
> $$H_{\text{val}} \triangleq \nabla_\theta^2 \, \mathbb{E}_{x \sim D_{\text{val}}} \left[ \ell(x; \theta) \right] \Big|_{\theta = \theta_s}.$$
>
> Let $H_{\text{val}} = V \Lambda V^\top$ be its spectral decomposition, where $\Lambda = \text{diag}(\lambda_1, \ldots, \lambda_d)$ with $\lambda_1 \geq \cdots \geq \lambda_d \geq 0$, and $V = [v_1, \ldots, v_d]$ is the orthonormal eigenbasis.

The eigenspectrum of $H_{\text{val}}$ captures the anisotropic curvature of the loss landscape. To formalize the distinction between high- and low-curvature directions, we partition the parameter space into two complementary subspaces using a curvature threshold $\epsilon > 0$. The stiff modes subspace consists of directions with eigenvalues greater than $\epsilon$, while the flat valleys subspace consists of directions with eigenvalues less than or equal to $\epsilon$. Formally, we introduce the following spectral partitioning:

> **Definition 4.2** (Spectral Subspace Partitioning). Given a curvature threshold $\epsilon > 0$, we partition the parameter directions into two disjoint subspaces:
>
> $$\mathcal{I}_\perp = \{j \mid \lambda_j > \epsilon\}, \quad \text{and} \quad \mathcal{S}_\perp = \text{span}\{v_j : j \in \mathcal{I}_\perp\},$$
> $$\mathcal{I}_\parallel = \{j \mid \lambda_j \leq \epsilon\}, \quad \text{and} \quad \mathcal{S}_\parallel = \text{span}\{v_j : j \in \mathcal{I}_\parallel\}.$$
>
> Here $\mathcal{S}_\perp$ denotes the stiff modes subspace (directions of high curvature) and $\mathcal{S}_\parallel$ the flat valleys subspace (directions of low curvature).

Based on the above partitioning, we connect the global geometry with sample-level gradient dynamics. For each training sample $x_i \in D_{\text{train}}$, we project its gradient at $\theta_s$ onto each eigenvector $v_j$.

> **Definition 4.3** (Sample-wise Spectral Projections). For a training sample $x_i$, we define its gradient projection onto $v_j$ at $\theta_s$ as: $g_{i,j} \triangleq \langle \nabla \ell_{\text{pre}}(x_i; \theta_s), v_j \rangle$.

We now formulate sample selection as an optimization problem that explicitly encodes the anisotropic curvature of the loss landscape. Our objective is to select samples that collectively produce a large gradient component in the flat subspace $\mathcal{S}_\parallel$ (to drive progress along low-curvature directions) while limiting the gradient energy in the stiff subspace $\mathcal{S}_\perp$ (to avoid instability). Formally, we propose the following optimization problem:

$$\max_{w \in \mathcal{W}} \left\| \sum_{i=1}^{N} w_i \boldsymbol{g}_{i,\mathcal{I}_\parallel} \right\|_2^2$$
$$\text{s.t.} \quad \sum_{j \in \mathcal{I}_\perp} \lambda_j \sum_{i=1}^{N} w_i g_{i,j}^2 \leq \tau_\perp, \tag{2}$$

where $\boldsymbol{g}_{i,\mathcal{I}_\parallel} = (g_{i,j})_{j \in \mathcal{I}_\parallel}$ is the projection of $\nabla \ell_{\text{pre}}(x_i; \theta_s)$ onto the flat subspace and $\tau_\perp > 0$ is a threshold controlling the allowed gradient energy in stiff directions. For data selection, we have $\mathcal{W} = \{w \in \{0,1\}^N \mid \sum_{i=1}^{N} w_i = K\}$, where $K$ is the number of samples to select.

### 4.3. Optimization Solver

The optimization problem in Equation (2) presents computational challenges due to its combinatorial nature for

---

**Algorithm 1** The Overall Pipeline of **DiReCT**

**Require:** Pre-trained model $\theta_s$, training set $D_{\text{train}}$, validation set $D_{\text{val}}$, selection count $K$ or reweighting flag
**Ensure:** Fine-tuned model $\theta_{\text{adj}}$
1: Generate random matrix $R \in \mathbb{R}^{k \times d}$ with $R_{ij} \sim \mathcal{N}(0, 1/k)$
2: **for** each $x \in D_{\text{val}}$ **do**
3:     Compute $z_x \leftarrow R\nabla_\theta \ell(x; \theta_s) \in \mathbb{R}^k$
4: **end for**
5: Form $\tilde{H} \leftarrow \frac{1}{M} \sum_{x \in D_{\text{val}}} z_x z_x^\top$
6: Compute $\tilde{H} = \tilde{V}\tilde{\Lambda}\tilde{V}^\top$ (eigendecomposition)
7: **for** each $x_i \in D_{\text{train}}$ **do**
8:     Compute $z_i \leftarrow R\nabla \ell_{\text{pre}}(x_i; \theta_s)$
9:     Set $g_{i,j} \leftarrow \sqrt{k} \cdot z_i^\top \tilde{v}_j$ for $j = 1, \ldots, k$
10: **end for**
11: Partition $\{1, \ldots, k\}$ into $\mathcal{I}_\perp$ and $\mathcal{I}_\parallel$ using threshold $\epsilon$
12: Solve Eq. (2) via Algorithm 2 to obtain weights $w$
13: Fine-tune $\theta_s$ with weighted loss $\mathcal{L}_{\text{pre}}(\theta; w)$ for a few steps $\rightarrow \theta_{\text{adj}}$
14: **return** $\theta_{\text{adj}}$

---

selection ($w_i \in \{0, 1\}$) and non-convex structure. For data selection, Equation (2) reduces to a cardinality-constrained binary quadratic program, which is NP-hard and intractable for typical pretraining dataset sizes ($N \sim 10^5 - 10^6$). For data reweighting ($w_i \geq 0$), the maximization of a convex quadratic function over a convex set remains non-convex. We therefore adopt a two-stage approach: first solve a continuous relaxation, then recover discrete solutions for selection through rounding. We begin by relaxing the binary constraint to $w_i \in [0, 1]$. Then we employ a Successive Convex Approximation (SCA) solver. The key idea is to iteratively construct and solve convex approximations of the original problem. At iteration $t$ with current solution $w^{(t)}$, we use the fact that for any $w, w^{(t)}$:

$$\|Gw\|_2^2 \geq 2(Gw^{(t)})^\top(Gw) - \|Gw^{(t)}\|_2^2,$$

where $G$ is the $|\mathcal{I}_\parallel| \times N$ matrix with columns $\boldsymbol{g}_{i,\mathcal{I}_\parallel}$. This inequality follows from the convexity of $\|Gw\|_2^2$ in $w$. The right-hand side provides a linear minorizer that touches the original objective at $w^{(t)}$. We thus solve the convex approximation:

$$\max_{w \in [0,1]^N} \quad 2(Gw^{(t)})^\top(Gw)$$
$$\text{s.t.} \quad \sum_{j \in \mathcal{I}_\perp} \lambda_j \sum_{i=1}^{N} w_i g_{i,j}^2 \leq \tau_\perp, \quad \sum_{i=1}^{N} w_i = K. \tag{3}$$

Equation (3) is a linear program with a convex quadratic constraint. We solve it efficiently using projected gradient

descent with Dykstra's projection algorithm for intersecting convex sets, which handles the combination of box constraints ($w_i \in [0, 1]$), equality constraint ($\sum_i w_i = K$), and quadratic constraint. The complete algorithm proceeds as follows. We initialize $w^{(0)} = (K/N, \ldots, K/N)$ and iterate until convergence (defined as $\|w^{(t+1)} - w^{(t)}\|_2 < \delta$). For data selection, we apply deterministic rounding: sort $w_i^*$ in descending order and select the top $K$ samples. Optionally, we can use randomized rounding with probability proportional to $w_i^*$, which provides approximation guarantees while maintaining the cardinality constraint exactly.

### 4.4. Implementation Details of DiReCT

**Hessian Approximation via Randomized Sketching.** Direct eigendecomposition of the full Hessian $H_{\text{val}} \in \mathbb{R}^{d \times d}$ is computationally prohibitive for large language models ($d \sim 10^9$). We instead approximate its dominant spectral structure using randomized sketching. Let $k \ll d$ be a target subspace dimension. We construct a random projection matrix $R \in \mathbb{R}^{k \times d}$ with i.i.d. $\mathcal{N}(0, 1/k)$ entries, set $z_j := R\nabla_\theta \ell(x_j^{\text{val}}; \theta_s) \in \mathbb{R}^k$, and compute the sketched curvature matrix

$$\tilde{H} \triangleq \frac{1}{M} \sum_{j=1}^{M} z_j z_j^\top \in \mathbb{R}^{k \times k}.$$

Its eigendecomposition $\tilde{H} = \tilde{V}\tilde{\Lambda}\tilde{V}^\top$ yields approximate eigenvalues $\lambda_j \approx \tilde{\lambda}_j$ and eigenvectors $v_j \approx \sqrt{k}R^\top \tilde{v}_j$. These define the spectral partitioning $\mathcal{I}_\perp$ and $\mathcal{I}_\parallel$ according to Definition 4.2. Gradient projections are computed efficiently as $g_{i,j} \approx \sqrt{k}\langle R\nabla\ell_{\text{pre}}(x_i; \theta_s), \tilde{v}_j \rangle$, reducing storage from $O(Nd)$ to $O(Nk)$.

**The Overall Pipeline.** Algorithm 1 outlines the complete **DiReCT** procedure. Given a pre-trained model $\theta_s$, we first approximate the Hessian eigenspace using randomized sketching: validation gradients are projected onto a low-dimensional subspace to form a sketched covariance matrix $\tilde{H}$, whose eigen-decomposition yields approximate curvature directions $\tilde{v}_j$ and eigenvalues $\tilde{\lambda}_j$. We then project each training-sample gradient onto these directions to obtain coefficients $g_{i,j}$. Using a curvature threshold $\epsilon$, the directions are partitioned into stiff ($\mathcal{I}_\perp$) and flat ($\mathcal{I}_\parallel$) subspaces. With these projections, the optimization problem (Eq. (2)) is solved via SCA to obtain sample weights $w^*$. Finally, starting from $\theta_s$, a few fine-tuning steps are performed using the weighted loss, yielding the adjusted model $\theta_{\text{adj}}$. The pipeline returns both the optimized weights and the fine-tuned model.

### 4.5. Theoretical Analysis

For the theoretical analysis we work with the regularized validation Hessian $H_\rho(\theta) \triangleq H_{\text{val}}(\theta) + \rho I$ for some small

$\rho > 0$ that ensures positive definiteness, and measure flatness around the late-phase parameter $\theta_s$ via the inverse-trace functional $S(\theta) = \text{Tr}(H_\rho(\theta)^{-1}) = \sum_{i=1}^{d} 1/\lambda_i(\theta)$, which is large precisely when $\theta$ sits in a flat valley of the loss landscape. Under mild spectral and Lipschitz regularity (formal assumptions in Appendix C), the surrogate $S$ exhibits a structural preference for flat eigendirections: moving along an eigenvector $v_k$ associated with a small eigenvalue $\lambda_k$ provably increases $S$ and contracts the corresponding curvature, as formalized below (proof in Appendix C).

**Theorem 4.4** (Flat-direction preference). *Under the spectral regularity assumptions of Appendix C,*

$$\partial_{v_k} S(\theta) \geq \frac{c_0}{2\lambda_k(\theta)^2} > 0. \tag{4}$$

*Consequently, for any sufficiently small $\eta > 0$, $S(\theta + \eta v_k) > S(\theta)$ and $\lambda_k(\theta + \eta v_k) < \lambda_k(\theta)$.*

Intuitively, flatness is exactly what a generalization bound charges for: PAC-Bayes guarantees penalize a model by the local curvature of the loss, so flatter basins mean smaller train–test gaps. Theorem 4.4 shows that a step along a flat direction flattens the landscape further, and Theorem C.5 (Appendix C) turns this gain into an explicit generalization guarantee with a shrinking curvature penalty. This is precisely what our objective (2) induces by concentrating mass on gradients in the flat subspace $\mathcal{S}_\parallel$, so the data **DiReCT** selects drives training toward flatter solutions that satisfy the generalization guarantee of Theorem C.5.

## 5. Experiments

In this section, we provide a comprehensive evaluation of our proposed approach. We begin by detailing the experimental setup in Section 5.1, covering the datasets, baselines, and technical configurations used in our study. We then report the main results in Section 5.2 to highlight the performance gains over existing methods. Finally, we provide practical guidance on hyperparameter selection in Section 5.3 and decipher the sample selection logic learned by **DiReCT** in Section 5.4 to interpret what kinds of data drive the observed improvements.

*Table 2.* Model architectures used in our experiments.

| | GPT-2-Medium | Llama-1.1B |
|---|---|---|
| Parameters | 355M | 1.1B |
| Layers | 36 | 22 |
| Attention Heads | 24 | 32 |
| Embedding Dim. | 768 | 2048 |
| Hidden Dim. | 3072 | 2048 |
| Max Seq. Length | 512 | 2048 |

*Table 3.* Composition of pre-training corpora by byte percentage. For The Pile, due to space limitations we report the top-6 sources and aggregate the remaining 16 specialized domains into a single "Others" entry.

| SlimPajama (for GPT-2-Medium) | | The Pile (for Llama-1.1B) | |
|---|---|---|---|
| **Data Source** | **Ratio (%)** | **Data Source** | **Ratio (%)** |
| Common Crawl | 54.10 | Pile-CC | 18.21 |
| C4 | 28.70 | PubMed Central | 14.48 |
| GitHub | 4.20 | Books3$^\dagger$ | 12.14 |
| Books | 3.70 | OpenWebText2$^\dagger$ | 10.07 |
| ArXiv | 3.40 | ArXiv | 9.01 |
| Wikipedia | 3.10 | GitHub | 7.63 |
| StackExchange | 2.80 | Others (16 sources) | 28.46 |

## 5.1. Experimental Setup

**Models and Pretraining Data.** To obtain the stable late-phase checkpoints $\theta_s$ essential for our curvature analysis, we pretrain multiple models across various scales and architectures (Xu et al., 2026). This selection is designed to investigate performance across distinct distributional characteristics. For the GPT series, we pretrain GPT-2-Medium on the SlimPajama dataset (Soboleva et al., 2023), a deduplicated and quality-filtered corpus derived from RedPajama. To verify the scalability of our method and to evaluate its adaptability to different model architectures, we also pretrain a Llama-1.1B model on The Pile (Gao et al., 2021), benefiting from its vast multi-domain exposure, particularly in specialized scientific and code sectors. This dual-model setup provides a distributionally diverse proxy for evaluating specialized reasoning tasks. The model architecture and dataset compositions are summarized in Table 2 and Table 3.

**Annealing Datasets.** To evaluate our proposed selection method during annealing, we construct a representative training mixture $D_{ann}$ by subsampling from three specialized corpora: MathPile (Wang et al., 2024) for mathematical reasoning, StarCoderData (Li et al., 2023) for code, and the Dolma 3 Longmino Mix (Soldaini et al., 2024) for long-context text. We preserve the relative proportions of the source corpora so that $D_{ann}$ remains a distributionally faithful proxy for specialized reasoning tasks while staying computationally tractable for the required second-order spectral analysis. Detailed statistics of the resulting mixture are provided in Appendix A.1.

**Benchmarks.** We evaluate our method on a suite of benchmarks spanning two key reasoning domains. For commonsense reasoning, we employ HellaSwag (Zellers et al., 2019), PiQA (Bisk et al., 2020), OpenBookQA (Mihaylov et al., 2018), COPA (Sarlin et al., 2020), ARC-Easy (Clark et al., 2018), and SciQ (Welbl et al., 2017) and WinoGrande (Sakaguchi et al., 2021). For mathematical and complex rea-

soning, we use GSM8K (Cobbe et al., 2021), HumanEval (Chen et al., 2021). All evaluations are reported with standard accuracy for most tasks and pass@1 for HumanEval.

**Baselines.** To rigorously evaluate the efficacy of **DiReCT**, we compare it against a suite of representative data selection baselines: Uniform Sampling, Perplexity-based selection (Hu et al., 2024), Loss-based selection, GradNorm (IS) (Katharopoulos & Fleuret, 2018), and InfoBatch (Qin et al., 2024). To ensure a fair comparison, we enforce a strict data budget across all methods by fixing the selection ratio at 80% of the original meta-dataset. This controlled experimental design ensures that the performance variations observed in Table 4 are attributable solely to the quality of the selection logic rather than the volume of training data.

## 5.2. Main Results

The experimental results in Table 4 yield three consistent findings across both model scales. First, **DiReCT** attains the highest aggregated score at both scales, gaining roughly two points over Uniform Sampling. This confirms that the improvements generalize across model capacity rather than relying on scale-specific tuning. Second, on the math and code reasoning benchmarks (GSM8K and HumanEval), **DiReCT** delivers clear gains that grow with model capacity, consistent with the intuition that larger models possess the latent reasoning ability that targeted data selection can unlock. Third, **DiReCT**'s effect on the commonsense reasoning benchmarks is heterogeneous: it lifts accuracy by several points on OBQA, SciQ, and ARC-E, while staying within roughly one point of Uniform Sampling on Hella., PiQA, COPA, and Wino. This pattern indicates that uniform sampling already suffices for surface-level linguistic patterns, whereas targeted gradient alignment is most valuable for navigating the geometric bottlenecks of deeper structural logic. The comparison with the strongest magnitude-centric baseline, InfoBatch, is particularly informative. InfoBatch matches or exceeds **DiReCT** on several individual commonsense benchmarks, yet **DiReCT** overtakes it on the aggregated metric at both scales. This indicates that maximizing "information density" is secondary to ensuring "update alignment." By projecting gradients onto the flat subspace of the validation Hessian, **DiReCT** biases the optimizer toward directions that generalize broadly, rather than toward sample difficulty in isolation. The small bidirectional fluctuations observed on commonsense tasks reflect a principled trade-off: imposing the stiff-mode constraint $\tau_\perp$ trades a fraction of the gradient energy on high-curvature directions in exchange for stable descent along the low-curvature valley floor, preserving foundational knowledge while aggressively specializing in the reasoning domains required for downstream applications.

*Table 4.* Performance comparison of **DiReCT** against various data selection baselines across two model scales (GPT-2-Medium 355M and Llama-1.1B) in the annealing stage. Subscripts denote the change relative to the Uniform Sampling baseline *at the same scale* (positive in red, negative in green). All numbers are averaged over 5 random seeds. **DiReCT** attains the highest aggregated score at both scales, with the most pronounced gains on the math and code reasoning benchmarks (GSM8K, HumanEval) and on OBQA, SciQ, and ARC-E.

| Method | Commonsense Reasoning | | | | | | | Math & Code Reasoning | | Aggregated |
|---|---|---|---|---|---|---|---|---|---|---|
| | Hella. | PiQA | OBQA | COPA | Wino. | SciQ | ARC-E | GSM8K | HumanE | Avg. |
| *GPT-2-Medium (355M)* | | | | | | | | | | |
| Uniform Sampling | 26.2 | 55.8 | 11.3 | 57.6 | 49.4 | 43.1 | 27.2 | 0.2 | 1.2 | 30.2 |
| Perplexity-based | 25.3 | 54.5 | 12.4 | 59.1 | 51.0 | 44.6 | 29.8 | 0.4 | 1.8 | 31.0 |
| Loss-based | 24.8 | 55.2 | 11.9 | 58.2 | 50.3 | 43.7 | 28.4 | 0.3 | 1.2 | 30.4 |
| GradNorm (IS) | 27.2 | 53.4 | 13.2 | 58.7 | 50.9 | 45.2 | 30.6 | 0.5 | 1.8 | 31.3 |
| InfoBatch | 27.5 | 56.8 | 12.7 | 59.4 | 51.5 | 46.0 | 31.7 | 0.7 | 3.0 | 32.1 |
| **DiReCT (Ours)** | $25.8_{-0.4}$ | $56.2_{+0.4}$ | $13.5_{+2.2}$ | $58.2_{+0.6}$ | $48.8_{-0.6}$ | $48.5_{+5.4}$ | $34.0_{+6.8}$ | $1.5_{+1.3}$ | $3.8_{+2.6}$ | $32.3_{+2.1}$ |
| *Llama-1.1B* | | | | | | | | | | |
| Uniform Sampling | 29.0 | 58.2 | 27.1 | 66.2 | 50.8 | 61.2 | 41.3 | 1.1 | 6.2 | 37.9 |
| Perplexity-based | 29.4 | 57.4 | 28.2 | 65.0 | 49.5 | 63.1 | 42.4 | 1.5 | 7.6 | 38.2 |
| Loss-based | 28.7 | 59.6 | 26.4 | 68.1 | 52.0 | 60.5 | 41.0 | 2.0 | 6.8 | 38.3 |
| GradNorm (IS) | 30.4 | 58.7 | 28.9 | 64.6 | 50.9 | 62.8 | 44.3 | 1.4 | 8.3 | 38.9 |
| InfoBatch | 31.2 | 60.5 | 28.4 | 69.3 | 52.4 | 65.6 | 43.7 | 2.5 | 7.1 | 40.1 |
| **DiReCT (Ours)** | $30.7_{+1.7}$ | $57.5_{-0.7}$ | $28.8_{+1.7}$ | $65.1_{-1.1}$ | $51.2_{+0.4}$ | $66.5_{+5.3}$ | $47.0_{+5.7}$ | $4.5_{+3.4}$ | $11.2_{+5.0}$ | $40.3_{+2.4}$ |

## 5.3. Practical Guidance for Hyperparameter Decision

A primary challenge in deploying **DiReCT** involves selecting the curvature threshold $\epsilon$, which defines the boundary between structural preservation and task adaptation. Rather than exhaustive grid searching, we propose a heuristic-based decision process grounded in the model's Hessian spectral dynamics. Importantly, we observe that the gradient energy budget $\tau_\perp$ is intrinsically coupled with the number of selected components $k$ and the required sample size for Hessian estimation. By modulating $\tau_\perp$, practitioners can identify the optimal $K$ that captures the dominant structural constraints while maintaining computational tractability.

The selection of $\epsilon$ is driven by the intrinsic dimensionality of the Hessian spectrum $H_{\text{val}}$. As illustrated in Figure 2(a), the eigenvalue distribution for GPT-2-Medium exhibits a characteristic power-law decay, suggesting that model sensitivity is concentrated within a remarkably low-dimensional manifold. To systematically decouple the parameter space, we employ a spectral elbow detection heuristic quantified by the cumulative energy $E(k) = (\sum_{i=1}^{k} \lambda_i)/(\sum_{j=1}^{D} \lambda_j)$.

Empirical results in Figure 2(c) demonstrate that the first $k = 34$ eigenvalues account for 94.5% of the total spectral energy. This "elbow" point serves as a geometric phase transition: the subspace $\mathcal{S}_\perp$ aligns with high-energy eigenvectors ($\lambda_i \geq \epsilon$) that constitute the steep, high-curvature walls of the loss landscape. These directions represent the rigid structural foundations of the pre-trained model, where any significant weight update would lead to catastrophic for-

getting. Conversely, the remaining spectral energy resides in the flat subspace $\mathcal{S}_\parallel$ ($\lambda_i < \epsilon$), corresponding to the valley floor. The diffused nature of these low-energy eigenvectors in Figure 2(b) confirms that $\mathcal{S}_\parallel$ provides the necessary plasticity for adaptive fine-tuning. By tuning $\tau_\perp$ to reach this elbow, we ensure that the optimization trajectory is orthogonal to the stiffest constraints, thereby stabilizing the learning process without sacrificing the model's capacity for new task acquisition.

## 5.4. Deciphering the Selection Logic

To understand the sample selection tendencies of **DiReCT**, we examine its behavior on two types of extreme samples: those with very high pre-training loss and those with very short sequence length. Figure 3(a) shows that selected samples exhibit markedly higher loss than those not selected, with the distribution centered significantly higher. This aligns with the **DiReCT** objective of maximizing projections onto the flat subspace $\mathcal{S}_\parallel$. During the late-phase training regime, low-loss samples typically provide weak gradient signals, indicating that the model has already converged in those parameter directions. In contrast, high-loss samples represent unlearned patterns that supply the gradient energy needed to drive optimization along the low-curvature "flat valleys." The strong correlation (Pearson $r = +0.811$) confirms that the algorithm effectively identifies samples with higher optimization potential. Even when the absolute lengths in the candidate pool are modest, **DiReCT** consistently prefers the longer ones. Figure 3(b) shows that

**Hessian Eigenvalue and Eigenvector Analysis**

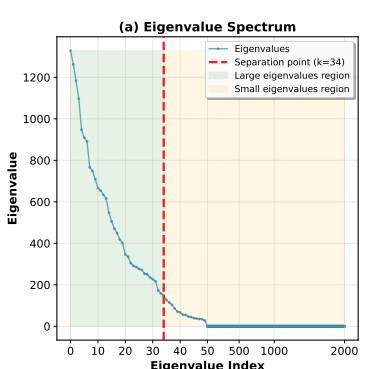
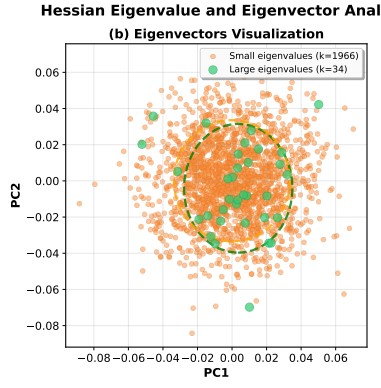
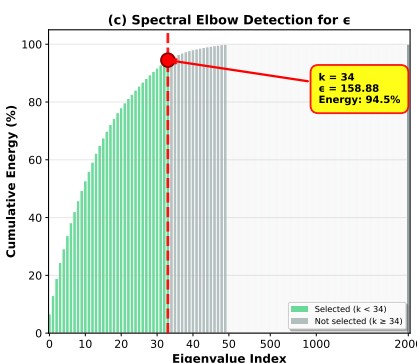

*Figure 2.* **Hessian Eigenvalue and Eigenvector Analysis for GPT-2-Medium.** (a) The eigenvalue spectrum exhibits a sharp power-law decay, where the green region denotes the stiff subspace and the orange region represents the flat subspace. (b) PCA projection of eigenvectors, illustrating the directional concentration of high-energy components versus the diffusion of low-energy ones. (c) Spectral elbow detection based on cumulative energy, identifying $k = 34$ as the optimal separation point ($\epsilon = 158.88$) covering 94.5% of the total spectral energy.

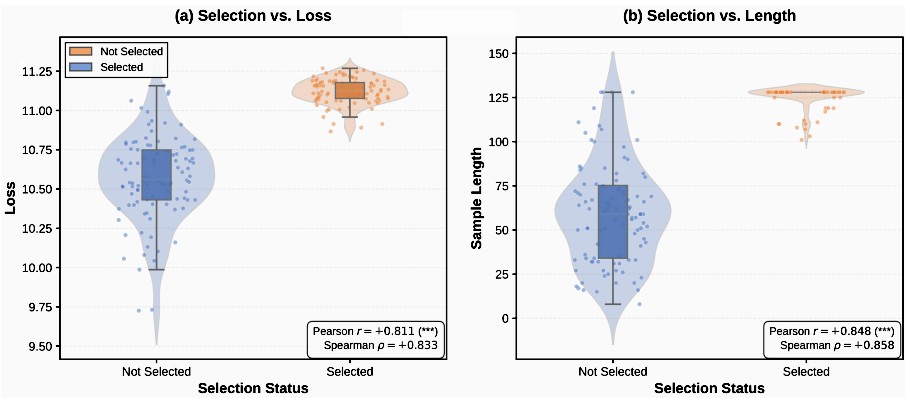

*Figure 3.* **Sample selection under two extreme regimes.** We probe **DiReCT**'s selection behavior on the constructed annealing dataset under two extremes: (a) the high-loss extreme, showing the pre-training loss distribution of selected versus unselected samples; and (b) the short-length extreme, showing the sequence length distribution. **DiReCT** prioritizes high-loss, long-sequence samples that align with the flat curvature of the loss landscape.

selected samples concentrate near 125 tokens, while shorter sequences are largely excluded. From a dynamical perspective, longer sequences often contain complex semantic dependencies and higher information density, yielding gradient vectors with richer components in the spectral decomposition that align with the flat directions of the validation Hessian $H_{\mathrm{val}}$.

# 6. Conclusion

We introduce **DiReCT**, a geometrically principled framework for annealing-phase data selection that replaces content-based heuristics with optimization-guided spectral analysis of the validation Hessian. The core idea is to up-weight samples aligned with the low-curvature valley floor and suppress high-curvature noise, which mitigates the convergence stagnation typical of late-stage pre-training while remaining scalable to foundation-model sizes. We further

give a PAC-Bayesian analysis showing that the induced flat-direction update contracts the curvature trace, yielding a generalization bound governed by this contracted trace. Empirically, **DiReCT** delivers consistent gains over standard baselines across diverse architectures.

# Limitations

**DiReCT** has two main limitations. First, the validation Hessian and gradient projections are computed only at a single onset-of-annealing checkpoint $\theta_s$; tracking the evolving landscape would sharpen the spectral partitioning but at substantial extra compute. Second, while randomized sketching reduces projection storage from $O(Nd)$ to $O(Nk)$, gathering gradients across the full $D_{\mathrm{train}}$ still requires a forward pass, which is nontrivial at trillion-token scale. We leave dynamic re-partitioning and parameter-subset backpropagation for gradient collection as future work.

## Impact Statement

This paper presents work whose goal is to advance the field of Machine Learning. There are many potential societal consequences of our work, none which we feel must be specifically highlighted here.

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

# A. Experiment Settings

## A.1. Annealing Datasets

We constructed a dataset for the annealing-phase experiments, comprising a 2.1-GB corpus of approximately 1.8M samples. To cultivate a multifaceted skill set, we curated data from three authoritative sources spanning distinct domains: math (1.27M samples from MathPile), code (0.23M samples from StarCoderData), and long-context text (0.30M samples from the Dolma 3 Longmino Mix). The dataset was partitioned into 1.62M training samples and 0.18M validation samples, following a 90/10 split. Figure 4 presents a detailed breakdown of the sample distribution and storage footprint across these domains.

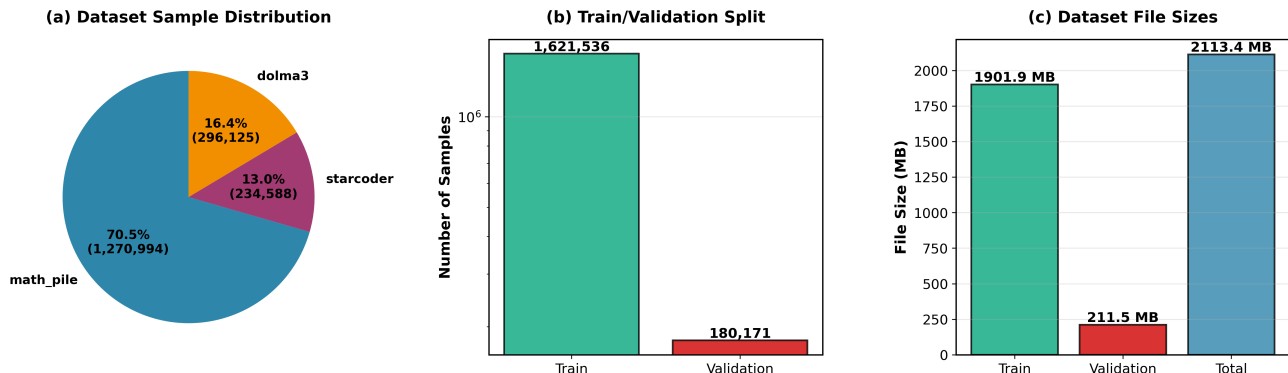

*Figure 4.* Overview of the annealing-phase dataset. (a) Sample distribution across sources: MathPile, StarCoderData, and the Dolma 3 Longmino Mix. (b) Train/validation split (90%/10%). (c) On-disk file sizes for train, validation, and total corpus.

## A.2. Implementation Details

**Baselines.** All baselines retain the top $80\%$ of the training set $D_{\text{train}}$ ranked by a method-specific score, except Uniform Sampling, which draws a random $80\%$ subset. The scoring metrics are Perplexity ($\exp \ell_{\text{pre}}$), Loss ($\ell_{\text{pre}}$ evaluated at the stable checkpoint $\theta_s$), and GradNorm ($\|\nabla_\theta \ell_{\text{pre}}\|_2$); higher-scored samples are kept. InfoBatch additionally rescales the kept gradients by $1.25$ to compensate for the pruned $20\%$.

**Hyperparameters.** The annealing phase uses $E = 5$ epochs and batch size $B = 4$. The learning rate is fixed at $\eta = 10^{-5}$ with weight decay $0.01$. The optimizer employs a cosine learning rate scheduler with warmup ratio $\rho = 0.1$, and gradient clipping at a maximum norm of $1.0$. For the DiReCT procedure, the sample weight vector $w$ is constrained by a gradient energy threshold $\tau_\perp = 0.1$ in the stiff subspace $\mathcal{S}_\perp$, under the nonnegativity constraint $w_i \geq 0$. The SCA solver (Algorithm 2) is run with maximum iteration count $T_{\max} = 20$ and convergence tolerance $\delta = 10^{-4}$.

# B. Successive Convex Approximation Solver

Our goal is to learn a continuous selection vector $w \in [0, 1]^N$ such that the selected samples produce a strong aggregate gradient in the flat subspace while remaining within a budget in the stiff subspace. Because the flat-subspace objective is quadratic in $w$, we adopt a Successive Convex Approximation (SCA) strategy: at each iteration we replace the quadratic objective with a linear surrogate around the current solution, solve the resulting linear program (LP), and update $w$.

**Original problem.** Let $G = [\boldsymbol{g}_{1,\mathcal{I}_\parallel}, \ldots, \boldsymbol{g}_{N,\mathcal{I}_\parallel}] \in \mathbb{R}^{|\mathcal{I}_\parallel| \times N}$ collect the per-sample gradients projected onto the flat subspace, and let $\boldsymbol{p}(w) = Gw$ denote the resulting aggregate gradient. The continuous relaxation of Equation (2) reads

$$\max_{w \in [0,1]^N} \|Gw\|_2^2 \quad \text{s.t.} \quad \sum_{j \in \mathcal{I}_\perp} \lambda_j \sum_{i=1}^N w_i g_{i,j}^2 \leq \tau_\perp, \quad \mathbf{1}^\top w = K.$$

The feasible set is convex (box, equality, and linear budget), but the objective $f(w) = \|Gw\|_2^2 = w^\top G^\top G w$ is quadratic in $w$, making direct maximization non-trivial.

**SCA surrogate.** At iteration $t$ with current solution $w^{(t)}$ and aggregate gradient $\boldsymbol{p}^{(t)} = Gw^{(t)}$, the gradient $\nabla f(w^{(t)}) = 2G^\top Gw^{(t)}$ yields the first-order minorizer $f(w) \geq f(w^{(t)}) + 2(Gw^{(t)})^\top G(w - w^{(t)})$. Discarding terms independent of $w$ and defining the per-sample marginal contribution and stiff-budget cost

$$c_i^{(t)} := 2\boldsymbol{p}^{(t)^\top} \boldsymbol{g}_{i,\mathcal{I}_\parallel}, \qquad a_i := \sum_{j \in \mathcal{I}_\perp} \lambda_j g_{i,j}^2,$$

the surrogate collapses into a linear program in $w$:

$$w^{(t+1)} = \arg \max_{w \in [0,1]^N} c^{(t)^\top} w \quad \text{s.t.} \quad a^\top w \leq \tau_\perp, \quad \mathbf{1}^\top w = K.$$

Intuitively, each SCA iteration is a re-scoring step: samples whose gradients are well-aligned with the current aggregate direction (large $c_i^{(t)}$) are favored, while those consuming excessive stiff-budget (large $a_i$) are suppressed.

**Solving the LP via projected gradient ascent.** We solve the LP with projected gradient ascent: each inner step takes $\tilde{w}^{(s+1)} = w^{(s)} + \eta c^{(t)}$ and is followed by an Euclidean projection $w^{(s+1)} = \Pi_{\mathcal{C}}(\tilde{w}^{(s+1)})$ onto the feasible set $\mathcal{C} = \{w : 0 \leq w_i \leq 1, \mathbf{1}^\top w = K, a^\top w \leq \tau_\perp\}$. Since $\mathcal{C}$ is the intersection of three simple convex sets, we compute $\Pi_{\mathcal{C}}$ via Dykstra's alternating projection. The outer SCA loop terminates once $\|w^{(t+1)} - w^{(t)}\|_2 < \delta$, returning the optimized weights $w^*$.

---

**Algorithm 2** SCA Solver

---

**Require:** Gradient matrix $G \in \mathbb{R}^{|\mathcal{I}_\parallel| \times N}$, per-sample squared projections $\{g_{i,j}^2\}_{i,j}$ for $j \in \mathcal{I}_\perp$, eigenvalues $\{\lambda_j\}_{j \in \mathcal{I}_\perp}$, threshold $\tau_\perp$, selection count $K$, initial weights $w^{(0)} \in [0,1]^N$ (default: $w_i^{(0)} = K/N$), tolerance $\delta > 0$ (default: $10^{-4}$), max iterations $T_{\max}$

**Ensure:** Optimized continuous weights $w^* \in [0,1]^N$

  1: $t \leftarrow 0$
  2: **repeat**
  3:     Compute current aggregate gradient: $\boldsymbol{p}^{(t)} \leftarrow Gw^{(t)}$
  4:     Form linear objective: $c_i \leftarrow 2\boldsymbol{p}^{(t)\top} \boldsymbol{g}_{i,\mathcal{I}_\parallel}$ for $i = 1, \ldots, N$
  5:     Solve convex subproblem:
  6:     $w^{(t+1)} \leftarrow \arg\max_{w \in [0,1]^N} \quad \sum_{i=1}^N c_i w_i$
  7:       s.t.   $\sum_{j \in \mathcal{I}_\perp} \lambda_j \sum_{i=1}^N w_i g_{i,j}^2 \leq \tau_\perp, \quad \sum_{i=1}^N w_i = K$
       *// Subproblem solved via PGD with Dykstra's projection*
  8:     $t \leftarrow t + 1$
  9: **until** $\|w^{(t)} - w^{(t-1)}\|_2 < \delta$ or $t \geq T_{\max}$
10: $w^* \leftarrow w^{(t)}$
11: **return** $w^*$

---

## C. Theoretical Analysis

We establish a rigorous connection between the flatness-seeking behavior of the inverse-trace surrogate and a PAC-Bayesian generalization bound. The analysis proceeds in two steps. First, we show that under mild spectral regularity conditions, the gradient of the surrogate has a non-negligible component along the eigenvector of a sufficiently small Hessian eigenvalue. Second, we demonstrate that a parameter update along this direction contracts the dominant flatness term in the PAC-Bayes bound, yielding a self-contained generalization guarantee for the adjusted model whose leading factor is the contracted curvature trace.

### C.1. Setup and Assumptions

Let $L_{\mathcal{S}}(\theta)$ be the empirical loss and $L_{\mathcal{D}}(\theta)$ the population loss. For a fixed regularization constant $\rho > 0$ that ensures positive definiteness, define the regularized Hessian $H_\rho(\theta) = \nabla_\theta^2 L_{\mathcal{S}}(\theta) + \rho I$. Note that $H_\rho$ shares the same eigenbasis as $H_{\text{val}}$ up to the empirical/validation sampling error and the $\rho I$ shift, so a flat (resp. stiff) direction of $H_\rho$ corresponds to a flat

(resp. stiff) direction of $H_{\text{val}}$ used by the algorithm. The central object of study is the inverse-trace surrogate

$$S(\theta) = \text{Tr}\big(H_\rho(\theta)^{-1}\big) = \sum_{i=1}^{d} \frac{1}{\lambda_i(\theta)}, \tag{5}$$

where $\lambda_1(\theta) \geq \cdots \geq \lambda_d(\theta) > 0$ are the eigenvalues of $H_\rho(\theta)$ (in the same descending order as in Definition 4.1) and $\{v_i(\theta)\}_{i=1}^{d}$ the corresponding orthonormal eigenvectors. We omit the argument $\theta$ when no confusion arises. For a unit vector $u$, the directional derivative is $\partial_u f(\theta) = \langle \nabla_\theta f(\theta), u \rangle$.

**Assumption C.1** (Local spectral regularity). There exists a neighborhood $\mathcal{N}_\theta$ of $\theta$ such that $H_\rho$ is twice continuously differentiable and all eigenvalues are simple on $\mathcal{N}_\theta$. By analytic perturbation theory for simple eigenvalues (Kato, 1995; Stewart & Sun, 1990), each $\lambda_i$ and $v_i$ is then differentiable on $\mathcal{N}_\theta$, and for any unit vector $u$ we have $\lambda_i(\theta + tu) = \lambda_i(\theta) + t\, \partial_u \lambda_i(\theta) + O(t^2)$ as $t \to 0$.

**Assumption C.2** (Directional sensitivity of eigenvalues). There exist constants $c_0, c_1 > 0$ such that, for the eigenvector $v_k$ corresponding to a sufficiently small eigenvalue $\lambda_k$, the following holds:

$$\big|\partial_{v_k} \lambda_k(\theta)\big| \geq c_0, \tag{6}$$

$$\big|\partial_{v_k} \lambda_i(\theta)\big| \leq c_1 \qquad \text{for all } i \neq k. \tag{7}$$

The sign of $v_k$ is chosen such that $\partial_{v_k} \lambda_k(\theta) \leq -c_0 < 0$.

Condition (6) quantifies the intuition that moving along a flat direction significantly alters the associated curvature; the negative sign indicates that this direction initially leads to flatter regions. Condition (7) is a mild regularity requirement limiting the influence of $v_k$ on other eigenvalues.

**Assumption C.3** (Spectral gap control). With $M := \sum_{i \neq k} \lambda_i^{-2} < \infty$, assume

$$\lambda_k(\theta)^2 \leq \frac{c_0}{2c_1 M}. \tag{8}$$

This inequality ensures that the contribution of the small eigenvalue dominates the gradient of $S$, making the bias toward $v_k$ provably positive.

For the generalization analysis, we start from the standard PAC-Bayesian flatness bound (McAllester, 1999; Dziugaite & Roy, 2017; Neyshabur et al., 2017) and specialize it to the curvature trace used throughout this paper. Assume the loss is bounded in $[0, 1]$. With probability at least $1 - \delta$ over the training sample, for all $\vartheta \in \mathcal{N}_\theta$, we have

$$L_\mathcal{D}(\vartheta) \leq L_\mathcal{S}(\vartheta) + \Phi(\vartheta) + O\left(\frac{\log(n/\delta)}{n}\right), \tag{9}$$

where the flatness complexity term is

$$\Phi(\vartheta) := \sqrt{\frac{\text{Tr}(H_\rho(\vartheta))\|\vartheta\|^2 + \ln(2n/\delta)}{2n}}. \tag{10}$$

We obtain this curvature-trace form from the generic sharpness–KL bound as follows. Under a Gaussian perturbation $\nu \sim \mathcal{N}(0, \sigma^2 I)$, the expected sharpness equals $\frac{\sigma^2}{2}\text{Tr}(H_\rho(\vartheta))$ to second order (using $\nabla L_\mathcal{S}(\theta) = 0$) and the KL term equals $\|\vartheta\|^2/(2\sigma^2)$. The standard bound leaves these as two separate terms; our refinement is to tie the perturbation scale to the local curvature, choosing the curvature-matched variance $\sigma^2 = 1/(2\,\text{Tr}(H_\rho(\vartheta)))$. This collapses both terms into the single complexity factor $\text{Tr}(H_\rho(\vartheta))\|\vartheta\|^2$ inside the root, making the bound depend on the curvature trace that our update directly contracts.

To control higher-order effects when taking a small step along $v_k$, we impose a mild regularity condition on the loss Hessian.

**Assumption C.4** (Locally Lipschitz Hessian). On the neighborhood $\mathcal{N}_\theta$, the Hessian $\nabla^2 L_\mathcal{S}$ is Lipschitz continuous: there exists a constant $L_H > 0$ such that $\|\nabla^2 L_\mathcal{S}(\vartheta) - \nabla^2 L_\mathcal{S}(\vartheta')\|_{\text{op}} \leq L_H \|\vartheta - \vartheta'\|$ for all $\vartheta, \vartheta' \in \mathcal{N}_\theta$. Moreover, assume $\nabla L_\mathcal{S}(\theta) = 0$.

This standard condition implies that the operator norm of the Hessian is locally bounded, and it controls the changes in eigenvalues and parameter norm under small perturbations; all higher-order terms are of order $O(\|\theta' - \theta\|^2)$. Moreover, by Assumption C.1, the trace $T(\vartheta) := \mathrm{Tr}(H_\rho(\vartheta))$ is twice continuously differentiable on $\mathcal{N}_\theta$, so we may set $L_T := \sup_{\vartheta \in \mathcal{N}_\theta, \|u\|=1} |\partial_u^2 T(\vartheta)| < \infty$; this $L_T$ controls the second-order Taylor remainder of $T$ along any unit direction.

## C.2. Gradient Bias Toward the Flat Direction

The following result, stated as Theorem 4.4 in the main text, formalizes the preference of the surrogate $S$ for the flat eigendirection $v_k$.

**Proof of Theorem 4.4.** Differentiating (5) along $v_k$ gives $\partial_{v_k} S(\theta) = -\partial_{v_k}\lambda_k/\lambda_k^2 - \sum_{i \neq k} \partial_{v_k}\lambda_i/\lambda_i^2$. By Assumption C.2, $-\partial_{v_k}\lambda_k \geq c_0$, so the first term is at least $c_0/\lambda_k^2$; the remaining terms are bounded using (7) as $\left|\sum_{i \neq k} \partial_{v_k}\lambda_i/\lambda_i^2\right| \leq c_1 \sum_{i \neq k} \lambda_i^{-2} = c_1 M$. Hence $\partial_{v_k} S(\theta) \geq c_0/\lambda_k^2 - c_1 M$, and applying the gap condition (8), $c_1 M \leq c_0/(2\lambda_k^2)$, we obtain (4). The monotonicity claims follow from first-order Taylor expansions, since the positive linear term dominates for small $\eta$.

## C.3. A PAC-Bayesian Generalization Bound

**Theorem C.5** (PAC-Bayes generalization bound). *Under the regularity assumptions of this appendix, there exist a threshold $\eta_0 = \Theta(\lambda_k)$ and a constant $\Delta = \Omega(\eta) > 0$ such that, for every $\eta \leq \eta_0$, the update $\theta' = \theta + \eta v_k$ contracts the curvature trace, $\mathrm{Tr}(H_\rho(\theta')) \leq \mathrm{Tr}(H_\rho(\theta)) - \Delta$, and, with probability at least $1 - \delta$,*

$$L_\mathcal{D}(\theta') \leq L_\mathcal{S}(\theta) + \sqrt{\frac{(T-\Delta)\,\|\theta'\|^2 + \ln(2n/\delta)}{2n}} + \widetilde{O}(\lambda_k^2), \tag{11}$$

*where $T := \mathrm{Tr}(H_\rho(\theta))$, the norm satisfies $\|\theta'\|^2 \leq \|\theta\|^2 + O(\lambda_k)$, and $\widetilde{O}$ absorbs the standard $O(\log(n/\delta)/n)$ PAC-Bayes residual.*

**Proof of Theorem C.5.** Write $T := \mathrm{Tr}(H_\rho(\theta))$, $T' := \mathrm{Tr}(H_\rho(\theta'))$, $q := \|\theta\|^2$, $q' := \|\theta'\|^2$, and $\Sigma_\perp := \sum_{i \neq k} \partial_{v_k}\lambda_i(\theta)$.[1]

By Taylor expansion with remainder $|R_T| \leq \frac{L_T}{2}\eta^2$ (Asm. C.4), Asm. C.2 ($\partial_{v_k}\lambda_k \leq -c_0$), and $|\Sigma_\perp| \leq c_0/2$,

$$T' - T = \eta\,\partial_{v_k}\lambda_k + \eta\,\Sigma_\perp + R_T \leq -c_0\,\eta + \frac{c_0}{2}\eta + \frac{L_T}{2}\eta^2 = -\Delta, \tag{12}$$

with $\Delta = \frac{c_0}{2}\eta - \frac{L_T}{2}\eta^2 = \Omega(\eta) > 0$ for $\eta \leq \eta_0$.

Taylor at $\theta$ with $\nabla L_\mathcal{S}(\theta) = 0$ (Asm. C.4) and $B_L := \sup_{\mathcal{N}_\theta} \|\nabla^2 L_\mathcal{S}\|_{op} < \infty$ gives $L_\mathcal{S}(\theta') \leq L_\mathcal{S}(\theta) + \frac{B_L}{2}\eta^2 = L_\mathcal{S}(\theta) + O(\lambda_k^2)$, where the last step uses $\eta \leq \eta_0 = \Theta(\lambda_k)$.

Since $\|v_k\| = 1$, the Cauchy–Schwarz inequality $|\langle\theta, v_k\rangle| \leq \|\theta\|$ and $\eta \leq \eta_0 = \Theta(\lambda_k)$ yield

$$q' = \|\theta + \eta v_k\|^2 = q + 2\eta\langle\theta, v_k\rangle + \eta^2 \leq q + 2\eta\|\theta\| + \eta^2 = q + O(\lambda_k). \tag{13}$$

The linear-in-$\eta$ term is retained inside the flatness factor.

Since $\Phi$ is increasing in its trace argument, plugging (12) and the norm bound (13) into $\Phi$ gives

$$\Phi(\theta')^2 = \frac{T'q' + \ln(2n/\delta)}{2n} \leq \frac{(T-\Delta)\,q' + \ln(2n/\delta)}{2n}, \qquad q' \leq q + O(\lambda_k).$$

Substituting the empirical-loss bound and this flatness term into the PAC-Bayes bound (9), with $\widetilde{O}(\lambda_k^2)$ absorbing both the $O(\lambda_k^2)$ empirical-loss residual and the $O(\log(n/\delta)/n)$ PAC-Bayes term, yields (11). Because $\Delta = \Omega(\eta) > 0$, the leading flatness factor is the strictly contracted trace $T - \Delta < T$. $\square$

---

[1] The base point $\theta$ is fixed, so $\|\theta\|$, $\lambda_k$, $B_L$, $L_T$, and $c_0$ are all fixed finite quantities. Every $O(\cdot)$ below denotes a non-asymptotic bound with an explicit constant that may depend on these base-point quantities; it is taken with respect to the step size $\eta \leq \eta_0 = \Theta(\lambda_k)$. For instance, $\eta\|\theta\| \leq \eta_0\|\theta\| = (c\,\|\theta\|)\lambda_k = O(\lambda_k)$ since $\eta_0 = c\lambda_k$.

# D. Notations

| Notation | Description | Type |
|---|---|---|
| **Datasets** | | |
| $D_{\text{train}}$ | Training dataset of text sequences | Set |
| $N$ | Number of training samples | Scalar |
| $x_i$ | $i$-th training sample (text sequence) | Data point |
| $D_{\text{val}}$ | Validation dataset | Set |
| $M$ | Number of validation samples | Scalar |
| $x_j^{\text{val}}$ | $j$-th validation sample | Data point |
| $D_{\text{post}}$ | Supervised posttraining dataset | Set |
| $L$ | Number of posttraining samples | Scalar |
| $(x_k^{\text{post}}, y_k^{\text{post}})$ | $k$-th posttraining sample and label | Data point + label |
| **Models and Parameters** | | |
| $\theta$ | Model parameters | Vector in $\mathbb{R}^d$ |
| $d$ | Parameter dimension | Scalar |
| $\theta_{\text{base}}$ | Model from standard ERM training | Vector in $\mathbb{R}^d$ |
| $\theta_s$ | Model parameters at end of stable training | Vector in $\mathbb{R}^d$ |
| $\theta_{\text{adj}}(w)$ | Adjusted parameters after weighted training | Vector in $\mathbb{R}^d$ |
| $\theta_{\text{adj}}$ | Final adjusted model after **DiReCT** procedure | Vector in $\mathbb{R}^d$ |
| **Loss Functions** | | |
| $\ell_{\text{pre}}(x;\theta)$ | Self-supervised pretraining loss | Scalar function |
| $\ell_{\text{post}}(x,y;\theta)$ | Posttraining loss function | Scalar function |
| $\mathcal{L}_{\text{ERM}}(\theta)$ | Empirical risk: $\frac{1}{N}\sum_{i=1}^{N}\ell_{\text{pre}}(x_i;\theta)$ | Scalar function |
| $\mathcal{L}_{\text{pre}}(\theta;w)$ | Weighted training objective: $\frac{1}{\sum_i w_i}\sum_{i=1}^{N} w_i\ell_{\text{pre}}(x_i;\theta)$ | Scalar function |
| $\mathcal{L}_{\text{val}}(\theta)$ | Validation loss: $\frac{1}{M}\sum_{j=1}^{M}\ell_{\text{pre}}(x_j^{\text{val}};\theta)$ | Scalar function |
| $\mathcal{L}_{\text{post}}(\theta)$ | Posttraining loss: $\frac{1}{L}\sum_{k=1}^{L}\ell_{\text{post}}(x_k^{\text{post}}, y_k^{\text{post}};\theta)$ | Scalar function |
| **Optimization Variables and Constraints** | | |
| $w$ | Sample weight vector: $w = (w_1, \ldots, w_N)$ | Vector in $\mathcal{W}$ |
| $\mathcal{W}$ | Feasible set for weights (reweighting or selection) | Set |
| $K$ | Number of samples to select (cardinality constraint) | Scalar |
| $\tau_\perp$ | Threshold for gradient energy in stiff directions | Scalar |
| $t$ | Iteration index in SCA algorithm | Scalar |
| $w^{(t)}$ | Weight vector at iteration $t$ | Vector in $\mathbb{R}^N$ |
| **Hessian Analysis and Spectral Decomposition** | | |
| $H_{\text{val}}$ | Validation Hessian matrix: $\nabla_\theta^2 \mathbb{E}_{x\sim D_{\text{val}}}[\ell(x;\theta)]\|_{\theta=\theta_s}$ | Matrix in $\mathbb{R}^{d\times d}$ |
| $\lambda_j$ | $j$-th eigenvalue of $H_{\text{val}}$ ($\lambda_1 \geq \cdots \geq \lambda_d \geq 0$) | Scalar |
| $v_j$ | $j$-th eigenvector of $H_{\text{val}}$ | Vector in $\mathbb{R}^d$ |
| $V$ | Orthonormal eigenbasis: $V = [v_1, \ldots, v_d]$ | Matrix in $\mathbb{R}^{d\times d}$ |
| $\Lambda$ | Diagonal eigenvalue matrix: $\Lambda = \text{diag}(\lambda_1, \ldots, \lambda_d)$ | Diagonal matrix |
| $\epsilon$ | Curvature threshold for spectral partitioning | Scalar |
| $\mathcal{I}_\perp$ | Indices of high-curvature directions: $\{j \mid \lambda_j > \epsilon\}$ | Index set |
| $\mathcal{I}_\parallel$ | Indices of low-curvature directions: $\{j \mid \lambda_j \leq \epsilon\}$ | Index set |
| $\mathcal{S}_\perp$ | Stiff modes subspace (high curvature) | Subspace |
| $\mathcal{S}_\parallel$ | Flat valleys subspace (low curvature) | Subspace |
| $g_{i,j}$ | Sample gradient projection: $g_{i,j} = \langle \nabla\ell_{\text{pre}}(x_i;\theta_s), v_j\rangle$ | Scalar |
| $\boldsymbol{g}_{i,\mathcal{I}_\parallel}$ | Gradient projection onto flat subspace: $(g_{i,j})_{j\in\mathcal{I}_\parallel}$ | Vector in $\mathbb{R}^{|\mathcal{I}_\parallel|}$ |
| $G$ | Matrix of flat subspace gradients ($|\mathcal{I}_\parallel| \times N$) | Matrix |
| **Sketching and Approximation** | | |
| $k$ | Sketching subspace dimension ($k \ll d$) | Scalar |
| $R$ | Random projection matrix: $R \in \mathbb{R}^{k\times d}$ with $R_{ij} \sim \mathcal{N}(0, 1/k)$ | Matrix in $\mathbb{R}^{k\times d}$ |
| $\tilde{H}$ | Sketched curvature matrix: $\frac{1}{M}\sum_{j=1}^{M}(R\nabla_\theta\ell(x_j^{\text{val}};\theta_s))(R\nabla_\theta\ell(x_j^{\text{val}};\theta_s))^\top$ | Matrix in $\mathbb{R}^{k\times k}$ |
| $\tilde{V}$ | Eigenvectors of $\tilde{H}$ | Matrix in $\mathbb{R}^{k\times k}$ |
| $\tilde{\Lambda}$ | Eigenvalues of $\tilde{H}$ | Diagonal matrix |
| $\tilde{v}_j$ | $j$-th eigenvector of $\tilde{H}$ | Vector in $\mathbb{R}^k$ |
| $z_x$ | Sketched gradient for validation sample: $z_x = R\nabla_\theta\ell(x;\theta_s)$ | Vector in $\mathbb{R}^k$ |
| $z_i$ | Sketched gradient for training sample: $z_i = R\nabla\ell_{\text{pre}}(x_i;\theta_s)$ | Vector in $\mathbb{R}^k$ |

