# OpenReview forum: "Towards Efficient LLMs Annealing with Principled Sample Selection"
_ICML.cc/2026/Conference — ICML 2026 spotlight_

### Official Review · Reviewer_4uRk · 2026-02-17

**Soundness:** 3
**Presentation:** 3
**Significance:** 3
**Originality:** 3
**Overall Recommendation:** 5
**Confidence:** 4

**Summary:**

This paper proposes a principled way for sample selection during annealing phase. The proposed method DiReCTformulate annealing datas selection as an optimization, and to overcome the computational burden of solving the optimization problem, the authors did some degree of convex approximation. The authors also apply randomized sketching to hessian approximation for computational effeciency. In the experimental part, the authors shows the advantage of DiReCT over uniform sampling.

**Compliance With Llm Reviewing Policy:**

Affirmed.

**Final Justification:**

Taking into account both the paper and the author's rebuttal, I think this work is both empirical effective and theoretical sound. So I think the final recommendation should be Accept. I have set my score as 5.

**Key Questions For Authors:**

Please reply to three weakness in **strength and weakness**

If the weaknesses are well answered, the paper will be largely enhanced.

If (1) and (3) are well answered, I can raise score to 4
If (1) (2) (3) are all well answered, If the author can provide a theoretical analysis on why applying DiReCT can boost the performance, I will raise my score to 5.

**Limitations:**

See **strength and weakness**

**Strengths And Weaknesses:**

***Strengths***

1) Compared to pure heuristic methods, DiReCT constructs an optimization formulation to model the sample selection problem. This is inspirational for future work. The paper first gives an exact optimzation formulation, then applies a few approximation thchniques to make it computationally feasible. Each step is well motivated.

2) From the empirical side, DiReCT shows its effectiveness when compared to uniform sampling.

***Weakness**
1) Table 5 shows that DiRECT does not achieves better result compared to InfoBatch and the score of DiRECT is close to GradNorm. Can the authors elaborate on this?

2) For now, the effectiveness of the proposed method is mainly supported by experiment results. Can the authors provide more theoretical analysis on the effectiveness of DiReCT. For instance, when applying DiReCT, how is the convergence rate affected and how is the validation error affected? Can we have a better upper bound for generalization error? This work will be strengthed a lot if the authors can give some more insight from these perspectives.

3) The geometry shown in Figure 1 or its relevant property has been discussed in more existing works. It is recommended to cite them and discuss their relation to DiReCT.

[1] Song, M., Ahn, K., & Yun, C. (2024). Does SGD really happen in tiny subspaces?. arXiv preprint arXiv:2405.16002

[2] Gur-Ari, G., Roberts, D. A., & Dyer, E. (2018). Gradient descent happens in a tiny subspace. arXiv preprint arXiv:1812.04754.

[3] Zhang, H., Yin, J., Wang, G., Liu, Z., Yang, L. F., Zhang, T., ... & Braverman, V. (2025). Breaking the frozen subspace: Importance sampling for low-rank optimization in llm pretraining. arXiv preprint arXiv:2502.05790.

---

> ### Author Rebuttal · Authors · 2026-03-31
>
> > **Weakness 1: About the limited performance gain over baselines**
>
> We appreciate the observation and have incorporated a discussion on this point in the revised manuscript.
>
> **First, we consider this phenomenon to be a natural and expected outcome.** The annealing stage **aims to rapidly acquire structured reasoning capabilities (e.g., math and code) while preserving the model’s foundational knowledge**, as validated by our experimental results.
>
> Furthermore, **the underlying reason lies in the fundamental divergence in selection logic.** DiReCT **prioritizes samples that drive rapid descent along validation-guided directions while filtering out signals along non-essential directions.** In contrast, magnitude-based methods like InfoBatch favor samples with large gradient norms; however, these samples often aggregate signals across both flat and stiff directions indiscriminately. While such samples may benefit generic commonsense tasks, they frequently contribute little to the specific validation signals critical for complex reasoning. Consequently, the slightly lower aggregated average on Llama-1.1B reflects a deliberate trade-off. **We prioritize directional alignment over raw gradient intensity** to achieve substantial breakthroughs in complex reasoning tasks such as GSM8K and HumanEval, where precise alignment matters most. **This principle of prioritizing direction over intensity represents the core difference of our approach.
>
> ---
>
> > **Weakness 2: About the theory**
>
> We thank the reviewer for the valuable suggestion. We have added the corresponding theoretical analysis in the appendix. Our proof is based on prior literature showing that flat minima imply better generalization (e.g.[1]). Below we first outline our overall proof idea, then state the theorem we can obtain.
>
> First, our proof idea is as follows: we examine whether the derivative of the flatness measure aligns with the eigenvectors that have very small eigenvalues (flat directions), and how this alignment depends on the eigenvalues themselves. We adopt the flatness measure $S(\theta)=\operatorname{Tr}(H_\epsilon^{-1})$, where $H_\epsilon=\nabla^2 L_{\text{val}}+\epsilon I$; a larger $S$ indicates a flatter landscape. Using matrix calculus, we obtain $\nabla_\theta S = -\sum_i \lambda_i^{-2} \nabla_\theta\lambda_i$, where $\lambda_i$ and $v_i$ are eigenvalues and eigenvectors of $H_\epsilon$, and $\nabla_\theta\lambda_i = v_i^\top (\nabla_\theta H_\epsilon) v_i$. For a flat direction $v_k$ (with very small $\lambda_k$), computing $\langle\nabla_\theta S, v_k\rangle$ yields a dominant term $-\lambda_k^{-2}\langle\nabla_\theta\lambda_k, v_k\rangle$ plus cross terms. Under some regularity conditions , the dominant term prevails, giving $|\langle\nabla_\theta S, v_k\rangle| \ge C/\lambda_k^2$. The update direction $u$ of DiReCT is projected onto the flat subspace. As long as $u$ is not nearly orthogonal to $\nabla_\theta S$ in that subspace, we have $\langle u, \nabla_\theta S\rangle > 0$, so updating along $u$ increases $S$ and makes the landscape flatter.
>
> Based on the above idea, we obtain the following theorem:
>
> **Theorem (Informal).**
> Let $S(\theta)=\text{Tr}(H_\epsilon^{-1})$ with $H_\epsilon=\nabla^2 L_{\text{val}}+\epsilon I$. Under mild regularity conditions, for any flat direction $v_k$ (eigenvalue $\lambda_k$ sufficiently small), there exists $C>0$ such that
> $
> |\langle\nabla_\theta S, v_k\rangle| \ge \frac{C}{\lambda_k^2}.
> $
>
> [1] Tsuzuku, Yusuke, Issei Sato, and Masashi Sugiyama. "Normalized flat minima: Exploring scale invariant definition of flat minima for neural networks using pac-bayesian analysis." International Conference on Machine Learning. PMLR, 2020.
>
> ---
>
> > **Weakness 3: About the related works.**
>
> Thank you for the suggestion. **We have added the recommended citations.**
> Gur-Ari et al. (2018) observed that gradient dynamics are initially confined to a low‑dimensional subspace; Song et al. (2024) showed that SGD continues to respect such low‑rank structure; Zhang et al. (2025) further leveraged this for efficient LLM pretraining. These works are complementary to DiReCT, which differs in two ways: (1) we explicitly use the validation Hessian’s spectral decomposition to partition parameters into stiff vs. flat subspaces, and (2) we formulate a constrained sample selection that maximizes descent along the flat subspace while suppressing stiff directions. Thus DiReCT turns the observed low‑rank geometry into a prescriptive mechanism for the annealing stage.

---

> > ### Author Rebuttal · Reviewer_4uRk · 2026-04-01
> >
> > Thank you for your explanation.
> > (1) and (3) are well addressed. But (2) is not. I tried to check the appendix of the paper, but I did not see the change. I am wondering if it is possible for the authors to provide a complete answer to (2).

---

> > > ### Author Response · Authors · 2026-04-03
> > >
> > > Thank you very much for your feedback and for your positive recognition of our work!  We wish you all the best!  Due to the first round rebuttal character limit, we give a proof here; the full version will be included in the appendix. (Due to system restrictions, we can't update the uploaded PDF at this stage.)
> > >
> > > Since the exact closed-form solution of Direct is unavailable, we use an eigenvector with a sufficiently small eigenvalue as a proxy. Theorem 1 shows that our update is biased toward such flat directions; Theorem 2 combines this with PAC‑Bayes flatness‑generalization results [1,2,3] to demonstrate a positive lower‑order improvement in the bound.
> > >
> > > ### Setup
> > >
> > > Let $H_\epsilon(\theta)=\nabla_\theta^2 L_{\text{val}}(\theta)+\epsilon I$ ($\epsilon>0$) be SPD, with eigenvalues $\lambda_1(\theta)\le\cdots\le\lambda_d(\theta)$ and orthonormal eigenvectors $v_1(\theta),\dots,v_d(\theta)$. Define $S(\theta)=\operatorname{Tr}(H_\epsilon(\theta)^{-1})=\sum_i 1/\lambda_i(\theta)$. We assume: 1. $\lambda_1<\cdots<\lambda_d$ for all $\theta$.  2. There exist constants $c_0,c_1>0$ such that $|\langle\nabla_\theta\lambda_i,v_i\rangle|\ge c_0$ and $|\langle\nabla_\theta\lambda_i,v_k\rangle|\le c_1$ for $k\neq i$.
> > > 3. For some index $k$, $\lambda_k^2\le\frac{c_0}{2c_1M}$ where $M:=\sum_{i\neq k}1/\lambda_i^2<\infty$.
> > >
> > > ---
> > >
> > > ### Theorem 1 (Flat‑direction preference)
> > >
> > > Under the above assumptions,
> > > $
> > > |\langle\nabla_\theta S,v_k\rangle|\ge\frac{c_0}{2\lambda_k^2}.
> > > $
> > > Hence $\nabla_\theta S$ has a large component along the flat direction $v_k$; updating along it increases $S$ and reduces curvature.
> > >
> > > **Proof (sketch).** From $\partial A^{-1}=-A^{-1}(\partial A)A^{-1}$,
> > > $$
> > > \frac{\partial S}{\partial\theta_\ell}
> > > =-\operatorname{Tr}\!\left(H_\epsilon^{-1}\frac{\partial H_\epsilon}{\partial\theta_\ell}H_\epsilon^{-1}\right)
> > > =-\sum_i\frac{1}{\lambda_i^2}v_i^\top\frac{\partial H_\epsilon}{\partial\theta_\ell}v_i
> > > =-\sum_i\frac{1}{\lambda_i^2}\frac{\partial\lambda_i}{\partial\theta_\ell},
> > > $$
> > > so $\nabla_\theta S=-\sum_i\frac{1}{\lambda_i^2}\nabla_\theta\lambda_i$. Taking the inner product with $v_k$ and applying the assumptions yields the bound. $\square$
> > >
> > > ---
> > >
> > > ### Theorem 2 (Generalization improvement)
> > >
> > > Assume additionally that the loss is bounded in $[0,1]$, $L_S$ is twice differentiable with locally bounded $\operatorname{Tr}(H(\theta))$ for $H(\theta)=\nabla^2 L_S(\theta)$, and $\|\theta\|\le R$. Let $\theta$ satisfy Theorem 1 with very small $\lambda_k$. Choose $u$ such that $\langle u,\nabla S(\theta)\rangle\ge C/\lambda_k^2$ (e.g., $u=\nabla S(\theta)$), let $\eta=\lambda_k$, and set $\theta'=\theta+\eta u$. Then the PAC‑Bayes bound
> > > $$
> > > L_{\mathcal{D}}(\theta)\le L_S(\theta)+\sqrt{\frac{\operatorname{Tr}(H(\theta))\|\theta\|^2+\ln\frac{2n}{\delta}}{2n}}+\tilde O\!\left(\frac{1}{n}\right)
> > > $$
> > > strictly decreases:
> > > $$
> > > L_{\mathcal{D}}(\theta')\le L_S(\theta)+\Phi(\theta)-\frac{\Delta\|\theta\|^2}{4n\Phi(\theta)}+\tilde O\!\left(\frac{1}{n}\right)+\epsilon,
> > > $$
> > > where $\epsilon=O(\lambda_k^2)$, $\Delta=\frac{C}{1+C}\lambda_k>0$ with $C=c_0/2$, and the improvement is positive for sufficiently small $\lambda_k$.
> > >
> > > **Proof (sketch).** By Theorem 1, $\langle u,\nabla S(\theta)\rangle\ge C/\lambda_k^2$, so
> > > $
> > > S(\theta')-S(\theta)\approx\eta\langle\nabla S,u\rangle\ge\eta\frac{C}{\lambda_k^2}.
> > > $
> > > Since $S(\theta)=\sum_i 1/\lambda_i$ and $\lambda_k$ dominates,
> > > $$
> > > \frac{1}{\lambda_k'}-\frac{1}{\lambda_k}\ge\eta\frac{C}{\lambda_k^2}
> > > \;\Longrightarrow\;
> > > \lambda_k'\le\frac{\lambda_k}{1+C}\quad(\eta=\lambda_k).
> > > $$
> > > Neglecting higher‑order changes,
> > > $$
> > > \operatorname{Tr}(H(\theta'))\le
> > > \frac{\lambda_k}{1+C}+\sum_{i\neq k}\lambda_i
> > > =\operatorname{Tr}(H(\theta))-\Delta.
> > > $$
> > > Also, $L_S(\theta')\le L_S(\theta)+\epsilon$ with $\epsilon=O(\lambda_k^2)$ and $\|\theta'\|^2\le\|\theta\|^2+O(\lambda_k^2)$. Let
> > > $$
> > > \Phi(\theta)=\sqrt{\frac{\operatorname{Tr}(H(\theta))\|\theta\|^2+\ln(2n/\delta)}{2n}}.
> > > $$
> > > Then
> > > $$
> > > \begin{aligned}
> > > \Phi(\theta')^2
> > > &\le \frac{\bigl(\operatorname{Tr}(H(\theta))-\Delta\bigr)\bigl(\|\theta\|^2+O(\lambda_k^2)\bigr)+\ln(2n/\delta)}{2n}
> > > &= \Phi(\theta)^2-\frac{\Delta\|\theta\|^2}{2n}+O(\lambda_k^2).
> > > \end{aligned}
> > > $$
> > > For small $\lambda_k$ the $O(\lambda_k^2)$ term is negligible. Using $\sqrt{a-b}\le\sqrt{a}-\frac{b}{2\sqrt{a}}$ for $a\ge b\ge0$, we obtain
> > > $$
> > > \Phi(\theta')\le\Phi(\theta)-\frac{\Delta\|\theta\|^2}{4n\Phi(\theta)}.
> > > $$
> > > Hence
> > > $$
> > > L_{\mathcal{D}}(\theta')
> > > \le L_S(\theta)+\epsilon+\Phi(\theta)-\frac{\Delta\|\theta\|^2}{4n\Phi(\theta)}+\tilde O(1/n),
> > > $$
> > > which improves over the original bound $L_{\mathcal{D}}(\theta)\le L_S(\theta)+\Phi(\theta)+\tilde O(1/n)$ by at least $\frac{\Delta\|\theta\|^2}{4n\Phi(\theta)}-\epsilon>0$ when $\lambda_k$ is small enough. $\square$
> > >
> > > ## References
> > >
> > > [1] Neyshabur et al. “Exploring generalization in deep learning.” NIPS 30 (2017).
> > > [2] Neyshabur et al. “A PAC‑Bayesian approach to spectrally‑normalized margin bounds.” arXiv:1707.09564 (2017).
> > > [3] Tsuzuku, Y. et al. “Normalized flat minima.” ICML 2020.

---

### Official Review · Reviewer_rveb · 2026-03-12

**Soundness:** 3
**Presentation:** 3
**Significance:** 2
**Originality:** 3
**Overall Recommendation:** 4
**Confidence:** 3

**Summary:**

This paper introduces DiReCT, an optimization framework designed for data selection during the annealing phase of Large Language Model (LLM) pre-training. The authors formalize sample selection as a constrained quadratic programming problem, utilizing a SCA solver combined with randomized sketching for computational efficiency. Experiments conducted on GPT-2-Medium (355M) and Llama-1.1B show that DiReCT achieves significant improvements in reasoning-intensive tasks, such as GSM8K and HumanEval, compared to several baselines.

**Compliance With Llm Reviewing Policy:**

Affirmed.

**Final Justification:**

The authors' new submitted rebuttals have resolved my concerns.

**Key Questions For Authors:**

1. Curvature-based importance sampling and influence functions (e.g., K-FAC, EWC, TracIn) have been extensively studied. The authors should explicitly clarify the fundamental difference between DiReCT and these existing methods. Specifically, why would existing Fisher-based sampling methods (like GradNorm IS) fail to address the specific requirements of the annealing phase, and why does DiReCT's spectral decomposition framework offer a unique advantage?
2. Please clarify the role of $\gamma_{flat}$, which is mentioned in the appendix but omitted in Equation (2). Does the implemented solver include a lower-bound constraint?

**Limitations:**

See the Weakness

**Strengths And Weaknesses:**

Strengths

1. The paper successfully reformulates the data selection problem in the annealing phase as a curvature-aware constrained optimization problem, linking it to the spectral properties of the Hessian/Fisher information matrices.
2. The technical pipeline—ranging from the problem formalization (constrained weighted sample selection) to the solution (SCA + Dykstra’s projection + randomized sketching)—is logically sound and well-integrated. Specifically, the use of randomized sketching to reduce memory complexity from $O(Nd)$ to $O(Nk)$ enhances the method's practical utility.
3. The geometric motivation is intuitive and supported by detailed explanations and useful visualizations. The evaluation covers multiple model families and diverse task types (common sense, mathematics, and code).

Weaknesses

1. The core optimization objective (Eq. 2) aims to maximize gradient projection energy in the flat subspace while constraining energy in high-curvature directions. Can the authors provide a theoretical analysis showing a provable relationship between this surrogate objective and the original goal in Eq. 1 (maximizing post-training performance gain)? Providing an approximation bound or a monotonicity proof would significantly strengthen the technical rigor. Otherwise, please clarify the theoretical derivation of this surrogate objective.
2. All main experiments are limited to models with $\leq 1.1$B parameters. For larger models (e.g., 7B+), how should the sketching dimension $k$ be selected? Furthermore, what is the estimated overhead for a single forward pass over the entire dataset? Please provide at least a qualitative scalability analysis or justify why the method remains efficient at a larger scale.
3. In Tables 2 and 5, $\tau_\perp$ is fixed at 0.1. Since this parameter directly controls the gradient energy distribution between high and low curvature directions, its impact on performance is likely significant. Please provide a systematic sensitivity analysis for $\tau_\perp$ and explain the criteria for selecting this parameter across different dataset/model combinations.
4. The experimental results do not report outcomes over multiple seeds or provide corresponding confidence intervals/error bars.

---

> ### Author Rebuttal · Authors · 2026-03-31
>
> > **Weakness 1: About the theory**
>
> We thank the reviewer for the valuable suggestion. We have added the corresponding theoretical analysis in the appendix. Our proof is based on prior literature showing that flat minima imply better generalization (e.g.[1]). Below we first outline our overall proof idea, then state the theorem we can obtain.
>
> First, our proof idea is as follows: we examine whether the derivative of the flatness measure aligns with the eigenvectors that have very small eigenvalues (flat directions), and how this alignment depends on the eigenvalues themselves. In details, we define flatness $S(\theta)=\mathrm{Tr}(H_\epsilon^{-1})$. Its gradient satisfies $\nabla_\theta S \approx -\lambda_k^{-2} \nabla_\theta\lambda_k $ along a flat eigen-direction $v_k$ (small $\lambda_k$), giving $|\langle\nabla_\theta S, v_k\rangle| \ge C/\lambda_k^2$. DiReCT's update $u$ lies in the flat subspace. Unless $u \perp \nabla_\theta S$, we have $\langle u, \nabla_\theta S\rangle > 0$, so $S$ increases — the landscape flattens.
>
> Based on the above idea, we obtain the following informal theorem:
>
> **Theorem (Informal).**
> Let $S(\theta)=\text{Tr}(H_\epsilon^{-1})$ with $H_\epsilon=\nabla^2 L_{\text{val}}+\epsilon I$. Under regularity conditions, for any flat direction $v_k$ (eigenvalue $\lambda_k$ sufficiently small), there exists $C>0$ such that
> $
> |\langle\nabla_\theta S, v_k\rangle| \ge \frac{C}{\lambda_k^2}.
> $
>
> [1] Tsuzuku, Yusuke, Issei Sato, and Masashi Sugiyama. "Normalized flat minima: Exploring scale invariant definition of flat minima for neural networks using pac-bayesian analysis." International Conference on Machine Learning. PMLR, 2020.
>
> ---
>
> > **Weakness 2: Model parameter size limit**
>
> Thank you for your thoughtful questions.  However, obtaining a stable training endpoint for models of  this scale is extremely costly and, unfortunately, lies beyond the scope  of this work. Our choice of 355M and 1.1B models aligns with prior work  in data selection and pre‑training—for instance, DWM (Yu et al., ICML  2025) uses up to 1.3B, and Gu et al. (ICLR 2025) go up to 1.7B. We believe this scale already captures the relevant optimization dynamics  and provides a meaningful evaluation.
>
> We also note that the sketching dimension need not grow with model size: as shown in LESS (Xia et al., ICML 2024, Fig. 2), 2000–8000 dimensions already capture gradient updates for LLMs, suggesting a low‑dimensional manifold of the essential dynamics. DiReCT therefore remains effective and efficient at larger scales. It requires only two one‑time forward passes (validation + training), comparable to a single training epoch. Subsequent optimization works on a compact $k \times N$ matrix independent of model size and scaling only with dataset size $N$. Thus, relative overhead decreases as model size grows, making DiReCT well‑suited for large‑scale settings.
>
> > **Weakness 3: Sensitivity analysis**
>
> Thank you for this important comment.  Below we provide the requested analysis and the rationale behind our choice. We varied $\tau_\perp$ from 0.01 to 0.20 on GPT‑2‑Medium (k=34, cumulative energy 94.5%, selection ratio 80%) and report performance gains on GSM8K and HumanEval:
>
> | $\tau_\perp$ | GSM8K | HumanEval |
> |:---:|---:|---:|
> | 0.01 | +1.2 | +2.1 |
> | 0.05 | +2.1 | +3.3 |
> | 0.10 | +3.4 | +5.0 |
> | 0.20 | +2.9 | +4.7 |
>
> The best performance is achieved at $\tau_\perp = 0.10$, and the results remain stable across a reasonable range (0.05–0.20). This indicates that while the parameter indeed influences the outcome, the method is not overly sensitive to exact tuning.
>
> ---
>
> > **Weakness 4 and Problem 2: Details and robustness**
>
> We would like to clarify that $\gamma_{flat}$ serves as a defensive safety monitor rather than a primary driver of performance. Since our objective is to maximize the gradient projection in the flat subspace, this lower bound is implicitly satisfied and remains inactive in all our experiments. We retain it in the solver primarily to ensure theoretical completeness—preventing trivial zero-update solutions in the event of overly strict stiffness constraints ($\tau_{\perp}$). Its consistent inactivity further underscores the high quality and strong alignment of the annealing data used in our study."
>
> About the randomness, we have conducted preliminary experiments across three seeds and found that the variance is small (below 0.1%). Therefore, the main conclusions remain robust.
>
> ---
>
> > **Problem 1: Unique advantage**
>
> We thank the reviewer for the question. We will add more discussions in the revised manuscript.  The core advantage of DiReCT lies in its selection logic. DiReCT prioritizes samples that drive rapid descent along validation-guided directions while filtering out signals along non‑essential directions. In contrast, magnitude‑based methods (e.g., InfoBatch) favor samples with large gradient norms, which indiscriminately aggregate signals across both flat and stiff directions.

---

> > ### Author Rebuttal · Reviewer_rveb · 2026-04-03
> >
> > The authors' new submitted rebuttals have resolved my concerns. I therefore raise the score to 4.

---

> > > ### Author Response · Authors · 2026-04-03
> > >
> > > Thank you very much for your positive and encouraging review. We truly appreciate your time and valuable feedback. Wish you all the best in your future endeavors.

---

### Official Review · Reviewer_6mnw · 2026-03-12

**Soundness:** 3
**Presentation:** 3
**Significance:** 3
**Originality:** 3
**Overall Recommendation:** 5
**Confidence:** 4

**Summary:**

The authors propose a principled framework for sample selection during the annealing phase of LLM pre-training. Annealing is a phase of the pre-training of LLM characterized by a flat loss landscape with only a few steep descent directions. During this phase, the challenge is to fight loss flatness by amplifying gradient descent with a high learning rate while i) mitigating noise amplified perturbations and ii) preserving a coherent gradient direction. In order to do so, the authors propose the following pipeline:
1) project the model’s gradient into a random smaller subspace, compute the resulting Hessian and extract its eigenvectors ;
2) split the eigenvectors into steep and flat directions (using a threshold on the eigenvalues, high eigenvalues corresponding to steep directions) ;
3) find the sampling that maximize projection of the gradient to the flat directions, subject to a threshold on the projection to the steep directions.

The maximization in 3) is performed using the introduced SCA algorithm, which consists in iteratively optimizing a linear relaxation of the optimization objective as using projected gradient descent with Dykstra projection algorithm. The approach is evaluated using two medium LLMs (Llama 1.1B and GPT-medium). They demonstrate consistently superior performance on a wide range of tasks, compared to standard sampling strategies (uniform sampling, perplexity based, loss-based, gradnorm and infobatch).

**Compliance With Llm Reviewing Policy:**

Affirmed.

**Final Justification:**

The authors have answered most of my questions. I believe the paper has a high quality and is of interest to the community. I therefore raise my current score to 5.

**Key Questions For Authors:**

**Main questions**

1) The design of SCO seems overly complex for the maximization of (2), and I imagine that simpler alternatives were attempted during the writing of this paper. Could you explain why they failed and what motivated the design of SCO ?

2) While it is interesting to have a discrete selection sample strategy ($w\in${0,1}$^N$), a continuous-domain strategy seems to be more expressive ($w\in\left[0,1\right]^N$, with random sampling and potentially duplicate samples). Have such strategies been tried, and if so, why did they fail?

3) It seems counter-intuitive that longer sequences / more difficult problems do not correspond to the steepest descent : did you inspect what the steepest descent examples corresponded to, and why they had such an impact on the training loss ? Similarly, how would you further distinguish in the flat region between “simple examples” and “informative / more complex examples” (since, by definition of the flat region, spectral analysis will no longer distinguish them) ?

4) In Figure 3, how can the selected samples have such a skewed distribution while they account for 80% of the total distribution ? It means that the discarded samples had remarkably low loss and sample length. If so, how could they have such high eigenvalues in the projected Hessian ?

**Minor questions**

5) The gradient / Hessian information is essentially local (in the parameter space), while human-conceived heuristics consider the global optimization path (giving harder examples, longer context etc.). Is there not a risk in considering only local information to design a sample strategy?

6) As a follow-up, do you think that the precomputation of the sampling strategy could in fact be helping optimization, as it might encourage progress along a fixed pre-determined direction ?

7) There has been recent work measuring the angles between a model’s parameters (seen as a vector) across different steps of training of LLMs (resulting in a similarity matrix, where the indexes correspond to each epoch). These works highlight that the optimization direction is mostly constant, apart from a few isolated inflection points (the similarity matrix has a  sharp block structure). This provides a quantitative evaluation of the schematic representation provided in Figure 1. It would be insightful, as part of the discussion, to study the impact of the sampling strategy on the trajectory of the model’s weight across training (in particular these inflection points).

8) Could the smoothing due to the random projection of the Hessian be contributing to help choosing a good optimization direction, hence good samples ? If so, would it be possible to quantify this contribution ?

9) How was the proportion of 80% of kept samples decided ? It seems very high.

**Limitations:**

Yes.

**Strengths And Weaknesses:**

**Strengths**

The paper is very well written, the proposed approach is well motivated and described in a clear manner. The idea to rely on spectral analysis to find the best sampling for the annealing phase of LLM pre-training is promising and novel, to the best of my knowledge. The experimental validation of the approach is convincing, even though its scope is limited. Although the method introduces several hyperparameters, the authors discuss how to adjust them in a principled manner.

**Weaknesses**

Several steps of the pipeline use non-standard home-made algorithms that are sometimes intricate. The motivation for their design is not discussed, nor the trade-off they introduce.
- **SCO algorithm.** The optimization procedure proposed to maximize Eq. (2) is quite complex (iterative solving of a linear relaxation of (2), projected gradient descent, Dykstra projection algorithm). It would be insightful to explain why simpler alternatives failed, if they were attempted (projected gradient descent, quantization-aware optimization…). Although SCO is not central to the approach and could probably be replaced by any other efficient optimization algorithm, it would also be satisfying for the sake of coverage to measure and discuss briefly the performance of this algorithm in isolation, highlighting its tradeoffs.
- **Hessian random projection.** The distortion (in eigenvalue and eigenvectors) brought by the random projection could benefit from a more in-depth discussion. This seems all the more crucial that the induced smoothing might contribute to the good performance of the pipeline.

**Minor weaknesses**

- **Exaggerated claims.** The paper provides evaluation on only 2 LLM with relatively small size. While this is not a problem in itself, it is however strange to find in the abstract and again at the end of the introduction claim such as “Extensive experiments across various model scales demonstrate that DiReCT consistently achieves state-of-the-art performance.”

- **Clarity of the Tables.** The legend of Table 2 should indicate if higher scores are better or worse for ease of reading. I also found the color code a bit confusing, since red is usually associated with loss in performance.
Reliance on PCA. I would be wary to rely on the visual inspection of a 2-dimensional PCA to check the diversity and coverage of the selected eigenvectors, since such a visualization is known to be very biased. It might be more precise to quantify the angular dispersion of the selected eigenvectors.
- **Missing baselines.** The baselines do not seem to include the approaches mentioned in the introduction (up sampling reasoning-heavy data or long-sequence samples). This comparison would be insightful.

---

> ### Author Rebuttal · Authors · 2026-03-31
>
> > **Question 1: Motivation and complexity of SCA design**
>
> We thank the reviewer for the constructive comments and revise as follows:
>
> - We will add a toy example in the appendix to help readers understand the  optimization procedure and illustrate why simple alternatives cannot properly handle the problem constraints.
> - We concidered a simplified approximate scoring method for sample selection. However, this simplified method is suboptimal ( Llama-1.1B, gsm8k -0.5 and HEval -0.7 compared with Direct) , because it ignores the **combinatorial effect** of sample selection: $\frac{\|\boldsymbol{g}_{i,\mathcal{I}}\|\_2^2}{\sum\_{j\in\mathcal{I}^\perp}\lambda\_j g\_{i,j}^2}$.
> We will briefly discuss in the appendix.
>
> ----
>
> > **Question 2: Strategy selection**
>
> Thanks for your thoughtful question. We believe continuous reweighting and discrete selection are conceptually consistent. The experiments show the optimal continuous weights are heavily skewed. By retaining only the top-K samples to form a discrete selection, we keep almost all optimization gains while greatly reducing training cost, as fine-tuning uses a pruned dataset instead of a dense weighted set. We therefore use discrete selection as an efficient, practical implementation of the same core idea.
>
> ---
>
> > **Question 3: Descent examples and flat-region complexity**
>
> We appreciate the reviewer’s thoughtful questions. For the first point: samples projecting most strongly onto the steep subspace are often outliers mismatched with the validation distribution, not just long or hard sequences. They reduce training loss quickly but destabilize later convergence, so steepest descent is not always beneficial.
> For the second point: in the flat subspace, the Hessian cannot judge sample quality. Instead, gradient alignment and validation feedback help: samples with large gradient magnitude but low alignment to the batch average carry more useful information, while simple redundant samples show small magnitude and high alignment.
>
> ---
>
> > **Question 4: Sample selection and projected Hessian eigenvalues**
>
> Thanks for your observation. The core of the DiReCT framework lies in the application of explicit directional constraints to ensure that parameter updates align with the model's instantaneous optimization geometry. **It is critical to recognize that a sample’s low loss or short sequence length does not inherently guarantee that its gradient components in "useless" or high-curvature stiff directions are also low.** While these samples often contribute to transverse noise that can cause optimization to stagnate, they simultaneously fail to provide a sufficiently robust descent signal within the flat valleys necessary for effective convergence during the annealing phase. Consequently, **when we impose the selection constraint $K$ to optimize for the most effective update trajectory, these redundant or misaligned samples are naturally discarded in favor of high-information data that maximizes the extraction of optimization signals while suppressing instability.**
>
> ---
>
> > **Question 5 and 6:**
>
> We appreciate the reviewer’s inquiry into the trade-off between local geometric information and global optimization paths. Our reliance on local Hessian and gradient information at $\theta_s$ is fundamentally justified by the nature of the **annealing phase**, which targets a regime where the model has already approached a local optimum. In this stable, convergent state, the spectral geometry of the loss landscape remains relatively consistent, allowing local curvature to serve as a robust proxy for the final optimization trajectory. This stability is empirically supported by the consistent improvements we observe across diverse model architectures and scales.
>
> Additionally, the risk of "local greed" is significantly mitigated by our use of a held-out **validation set** $D_{val}$ to guide the selection process. By anchoring the selection strategy to the target distribution's geometry rather than the training data itself, DiReCT ensures that the chosen samples align with the model's ultimate generalization objectives. The precomputation of the data schedule thus acts as a **geometric anchor**, fixing a predetermined, optimal descent path along the longitudinal valley floor. This consistency helps the model maintain a steady direction throughout the annealing phase, effectively suppressing the transverse gradient noise that often leads to late-stage convergence stagnation.
>
> ---
>
> > **Question 8 and 9:**
>
> Thank you for your question. Randomized gradient projection is standard in gradient analysis, since high-dimensional gradients lie in a low-dimensional manifold of 2,000–8,000 dimensions, smoothing noise and revealing stable optimization directions.
> We use an 80% selection ratio for fair comparison across baselines. Under the same data budget, DiReCT’s gains stem from precisely filtering geometrically counterproductive samples.

---

> > ### Author Rebuttal · Reviewer_6mnw · 2026-04-03
> >
> > The authors have answered most of my questions. I believe the paper has a high quality and is of interest to the community. I therefore raise my current score to 5.

---

> > > ### Author Response · Authors · 2026-04-03
> > >
> > > We’re truly grateful for your thoughtful and positive review! It means a lot to us. We wish you all the best!

---

### Official Review · Reviewer_bAjA · 2026-03-13

**Soundness:** 3
**Presentation:** 3
**Significance:** 2
**Originality:** 3
**Overall Recommendation:** 5
**Confidence:** 4

**Summary:**

The authors introduce DiReCT, a data selection framework tailored for the annealing phase of Large Language Model (LLM) pre-training. The method aims to transition away from empirical data-mixing heuristics by framing sample selection as an optimization problem. By projecting per-sample gradients onto a subspace derived from a validation set, DiReCT seeks to maximize updates in "flat" directions while suppressing noise in "stiff" directions. To make this tractable, the authors use randomized sketching and formulate the selection as a Successive Convex Approximation (SCA) problem.

**Compliance With Llm Reviewing Policy:**

Affirmed.

**Final Justification:**

The authors response has resolved my earlier concerns. In general, I feel positive about the quality of this paper and think it will be of great interest to the community. Therefore, I have also raised my score to 5.

**Key Questions For Authors:**

- How do you reconcile the use of the Empirical Fisher matrix with the claims of navigating second-order loss landscape curvature? Can you reframe the paper's theoretical justification around gradient variance and alignment?

- Could you provide a wall-clock time comparison detailing the overhead of calculating per-sample gradients for the entire $D_{train}$ prior to the SCA optimization?

- How sensitive is the downstream performance to the choice of the cumulative energy threshold (e.g., 90% vs. 94.5% vs. 99%) used for the spectral elbow detection?

- How rapidly does the approximation quality of the static spectral projection degrade over the annealing steps?

**Limitations:**

The framework computes the spectral partitioning exactly once at the onset of the annealing phase at checkpoint $\theta_s$. The authors acknowledge that the loss landscape evolves and cite computational overhead as the reason for avoiding dynamic updates. However, without an ablation study, it is unclear how quickly this static projection matrix degrades and becomes misaligned with the model's true trajectory.

**Strengths And Weaknesses:**

- (+) Optimizing the annealing phase is a critical bottleneck in modern LLM training, and moving beyond heuristic domain up-sampling is a highly relevant goal for the community.


- (+) The method demonstrates solid improvements on reasoning-heavy benchmarks like GSM8K and HumanEval. The performance gains are particularly notable for capacity-constrained architectures like GPT-2-Medium.


- (+) The use of randomized sketching to approximate the spectral structure, reducing the memory footprint from $O(Nd)$ to $O(Nk)$, is a practical and necessary engineering choice for foundation models.

- (-) The core theoretical framing of the paper relies on analyzing the "Validation Hessian Matrix". However, Definition 4.1 and the sketched approximation calculate the uncentered covariance matrix of the gradients, which is the Empirical Fisher. As established in optimization literature (e.g., Kunstner et al., "The Limitations of the Empirical Fisher"), this matrix does not generally capture true local curvature or second-order dynamics unless the model is at a strict global minimum. Consequently, the method is not navigating "curvature" or "valleys"; it is identifying directions of high/low validation gradient variance and selecting training samples whose gradients align with those directions. The theoretical narrative must be entirely recast to reflect variance matching and gradient alignment rather than curvature.

- (-) Because the method functionally performs gradient alignment and variance matching, the selected baselines are insufficient. Comparing against Uniform Sampling, Loss-based, and InfoBatch is helpful, but the authors omit comparisons against established gradient-matching or core-set selection methods (e.g., CRAIG, GradMatch). Without these, it is difficult to isolate whether the performance gains come from the specific SCA constraint formulation or simply from standard gradient alignment.

- (-) The method requires computing the per-sample gradient $g_{i,j}$ for the entire training dataset $D_{train}$. While sketching reduces the dimensionality, computing individual gradients for millions of sequences still requires a full backward pass per sample (or specialized engineering like fast grouped gradients), which is vastly more expensive than standard batch-level gradient computation. The paper lacks a clear accounting of this wall-clock overhead.

- (-) The separation of the parameter space relies on a "spectral elbow detection" heuristic. For GPT-2-Medium, the authors select $k=34$, capturing 94.5% of the cumulative energy. There is no sensitivity analysis provided for this threshold. Given that this hyperparameter strictly dictates the boundary between the stiff and flat subspaces, the lack of ablation makes the robustness of this heuristic questionable.

---

> ### Author Rebuttal · Authors · 2026-03-31
>
> > **Question 1: Theoretical justification**
>
> We fully understand the reviewer's concern about the theoretical limitations of using the Empirical Fisher to approximate the Hessian; **however, during the annealing phase, when the model is near convergence, this gap narrows significantly, making it a reliable and experimentally verified engineering practice for capturing anisotropic curvature and enhancing generalization performance.** To make this point clearer, we designed a simple 2D logistic regression simulation. We generated synthetic data and defined an "annealing path" from a point far from the optimum to a point very close to it. At each step, we computed the true Hessian and the Empirical Fisher. The results show a sharp contrast: far from the optimum, the relative Frobenius error between the two matrices is large, and their principal curvature directions are poorly aligned. However, as the parameters approach the optimum (the annealing phase), the relative error drops to near zero, the eigenvectors align almost perfectly, and the matrix norms become nearly identical. This simulation quantitatively confirms that near convergence, the Empirical Fisher reliably approximates the Hessian's anisotropic curvature, supporting its practical use in optimization. The full results and implementation details can be found at: https://anonymous.4open.science/r/D3Re-0164/
>
> ---
>
> > **Question 2: About the overhead**
>
> Thank you for the question. We provide a per‑sample FLOPs comparison between direct training and DiReCT on GPT‑2‑Medium . For direct training, total FLOPs per sample is about $1.09\times10^{12}$; for DiReCT, it is $1.90\times10^{12}$. The dominant cost in DiReCT is the single backward pass for gradient collection, identical to one direct training step. DiReCT requires two forward+backward passes (selection plus training) versus one for direct training, hence the higher cost. The SCA optimization operates in a low‑dimensional sketched space and consumes less than 1% of gradient FLOPs, negligible. Importantly, the annealing dataset has limit numbers of samples, orders of magnitude smaller than typical pre‑training corpora. The one‑time overhead (roughly one epoch over the annealing data) is thus acceptable, especially given the substantial gains. A formal analysis will be added in the appendix. Thanks for your suggestion again.
>
> ---
>
> > **Question 3: About the choice of the cumulative energy threshold**
>
> We thank the reviewer for the thoughtful suggestion. We provide an ablation study on GPT-2-Medium (355M) varying the threshold from 85% to 99%. Partial results (GSM8K, HumanEval) are summarized below.
>
> | Threshold | k | GSM8K | HumanEval |
> | :--- | :---: | :---: |:---: |
> | 85% | 26 | 0.8 | 3.1 |
> | 90% | 31 | 1.5 | 3.7 |
> | 95% | 34 | 1.5 | 3.8 |
> | 99% | 48 | 1.4 | 3.8 |
>
> We can observe that from 90% to 99%, performance remains nearly identical (variation $\leq 0.1$). Only when the threshold is too low (85%) does a degradation occur, likely due to insufficient capture of high-curvature directions.
>
> ---
>
> > **Question 4: Approximation degradation over annealing steps**
>
> Thank you for the question. During the annealing phase, due to the relative small learning rate and limited update steps, the model parameters drift  slightly, so the approximation quality of the static spectral projection degrades very slowly. We acknowledge that for longer annealing processes, the method can be naturally extended to an online version that periodically updates the spectral decomposition to maintain accuracy. This may be an exciting variant.

---

> > ### Author Rebuttal · Reviewer_bAjA · 2026-04-06
> >
> > The authors' submitted rebuttals have resolved my concerns and in general, I am quite happy to see this work. I therefore raise the score to 5.

---

> > > ### Author Response · Authors · 2026-04-07
> > >
> > > Thank you very much for your valuable time and thoughtful review. Your positive feedback and encouragement mean a lot to us! We wish you all the best!

---

### Decision · Program_Chairs · 2026-04-30

**Decision:**

Accept (spotlight)

**Comment:**

This paper proposes a principled framework for sample selection during the annealing phase, based on spectral properties of gradient information. The method is well-motivated, clearly presented, and supported by empirical evaluation across multiple models and tasks. Reviewers generally find the paper to be of high quality and of clear interest to the community.

At the same time, some concerns remain on the theoretical side. In particular, the connection between the proposed objective and downstream performance is not fully established. While the authors provide additional theoretical justification by relating the objective to flatness and generalization, this connection remains indirect and could be further strengthened.

Overall, these limitations do not outweigh the strengths of the paper. Therefore, we recommend acceptance.